# The impact of pathological high-frequency oscillations on hippocampal network activity in rats with chronic epilepsy

Laura A Ewell[1,2]*, Kyle B Fischer[1,3], Christian Leibold[4,5], Stefan Leutgeb[1,6], Jill K Leutgeb[1]*

[1]Neurobiology Section and Center for Neural Circuits and Behavior, Division of Biological Sciences, University of California, San Diego, La Jolla, United States; [2]Institute of Experimental Epileptology and Cognition Research, University of Bonn Medical Center, Bonn, Germany; [3]Neuroscience Graduate Program, University of California, San Diego, La Jolla, United States; [4]Department Biologie II, Ludwig-Maximilians-Universität München, Martinsried, Germany; [5]Berstein Center for Computational Neuroscience Munich, Martinried, Germany; [6]Kavli Institute for Brain and Mind, University of California, San Diego, La Jolla, United States

**Abstract** In epilepsy, brain networks generate pathological high-frequency oscillations (pHFOs) during interictal periods. To understand how pHFOs differ from normal oscillations in overlapping frequency bands and potentially perturb hippocampal processing, we performed high-density single unit and local field potential recordings from hippocampi of behaving rats with and without chronic epilepsy. In epileptic animals, we observed two types of co-occurring fast oscillations, which by comparison to control animals we could classify as 'ripple-like' or 'pHFO'. We compared their spectral characteristics, brain state dependence, and cellular participants. Strikingly, pHFO occurred irrespective of brain state, were associated with interictal spikes, engaged distinct subnetworks of principal neurons compared to ripple-like events, increased the sparsity of network activity, and initiated both general and immediate disruptions in spatial information coding. Taken together, our findings suggest that events that result in pHFOs have an immediate impact on memory processes, corroborating the need for proper classification of pHFOs to facilitate therapeutic interventions that selectively target pathological activity.

DOI: https://doi.org/10.7554/eLife.42148.001

*For correspondence:
laura.ewell@ukbonn.de (LAE);
jleutgeb@ucsd.edu (JKL)

**Competing interests:** The authors declare that no competing interests exist.

## Introduction

In epilepsy, transient fast oscillations in local field potentials called pathological high-frequency oscillations (pHFOs) are observed in brain regions near the seizure focus and are hypothesized to play a role in epileptogenesis (*Bragin et al., 2004*; *Staba et al., 2002*). In the clinical setting, interictal pHFOs are used as a biomarker for the location of seizure generating zones (*Worrell et al., 2004*) and are utilized by neurosurgeons to select which tissue to resect in surgical treatments of refractory epilepsy (*Haegelen et al., 2013*; *Frauscher et al., 2017*). The possible role of pHFOs in epileptogenesis and their use as a biomarker of disease highlight the necessity of being able to properly identify and classify physiological high frequency oscillations, which can be difficult in brain regions that also normally generate physiological high-frequency oscillations (*Engel et al., 2009*).

In temporal lobe epilepsy, seizure-genic circuits are located in brain regions that support memory formation such as the hippocampal and parahippocampal regions. In these regions, the healthy brain generates physiological high frequency oscillations or 'ripples', which are necessary for memory processing (*Axmacher et al., 2008*). Ripples overlap in frequency range with pHFOs, which presents

a challenge for algorithms to distinguish the two types of events (*Engel et al., 2009*). In human studies that have separately classified normal physiological ripples and pHFOs, the two event types were recorded from different electrode locations in segregated brain regions (*Matsumoto et al., 2013*), leaving the question of whether physiological ripples persist within seizure-genic networks that are producing pHFO, such that the same network generates both event types. Furthermore, it is difficult to know whether ripples observed in humans with epilepsy are normal or altered because there are no control data from healthy brains available and because of the difficulty in humans to simultaneously record the activity of many single neurons within a collective brain network. Here, we address these questions in an animal model of chronic temporal lobe epilepsy, in which we can record high-frequency oscillations and the neuron ensembles that participate in them from both epileptic and control rats.

Since the discovery of ripples in the rat hippocampus (*Buzsáki et al., 1992*), there has been extensive study of their mechanisms and function (for review, see *Buzsáki, 2015*). Ripples occur predominantly during sleep and immobility. Interestingly, the neuronal activation during ripples reflects a compressed version of the sequential activation of neurons during wakefulness (*Wilson and McNaughton, 1994*). When rats are awake and mobile, the hippocampal local field potential shows a theta rhythm, a sustained low amplitude oscillation that ranges between 5 and 12 Hz (*Vanderwolf, 1969*). During ongoing theta, individual principal neurons burst theta rhythmically when the rat is in the neuron's spatial receptive field (*O'Keefe and Dostrovsky, 1971*; *Muller et al., 1987*; *Ylinen et al., 1995*; *Csicsvari et al., 1999*). As a result of each neuron being theta-rhythmically active within spatially restricted fields, principal neurons are sequentially activated within theta cycles when animals transverse a space. These emerging theta sequences are then compressed and replayed during ripples, which is thought to be important for consolidation of memory during sleep as well as for decision-making and path planning during behavior (*Girardeau et al., 2009*; *Jadhav et al., 2012*; *Ólafsdóttir et al., 2015*; *Pfeiffer and Foster, 2013*). These direct links between behavior and neural network dynamics underscore the importance of ripple dynamics in healthy cognitive processes. In epilepsy, it is known that CA1 ensembles are activated during high frequency oscillations (*Ibarz et al., 2010*), while sequence coding is likely disrupted due to non-specific activation of pyramidal neurons (*Valero et al., 2017*). To date, however, no study has compared CA1 dynamics between pathological and more physiological (ripple-like) high-frequency oscillations in the same animal to test whether memory-related neural computations persist in seizure-genic networks. Using high-density single-unit recording, we compare the neural dynamics of high-frequency oscillations between control rats and those with chronic temporal lobe epilepsy. In animals with epilepsy, we find that the CA1 network of the same animal can generate pHFO and ripple-like oscillations. Here, we compare the brain state dependencies, cellular participation, and functional role of pHFO and ripple-like oscillations with control ripple oscillations.

## Results

### Ripple-like oscillations and pHFOs occur in CA1 of the same animal

We first determined whether the hippocampus of animals with chronic epilepsy is capable of engaging in two types of high-frequency oscillations, pathological and more normal, ripple-like events. Given that brain states differ between foraging and rest and that normal ripple oscillations are dependent on select brain states, we reasoned that recording across immobile and mobile behavioral states would be useful for assessing the pathology of high-frequency oscillations in epilepsy. Furthermore, recording during foraging behavior would also facilitate the study of the interplay between place coding and high-frequency oscillations in epilepsy. Recording sessions were thus comprised of two to four 10 min foraging epochs during which rats actively searched for food in an open arena, separated by rest periods during which rats quietly rested in a small box before and after foraging (*Figure 1A*) (n = 4 control, n = 4 with chronic epilepsy). For recording local field potential (LFP) and single units, tetrodes were positioned in the dorsal CA1 cell layer, where ripple oscillations are routinely recorded in healthy animals. Using identical event detection parameters, we isolated high-frequency oscillations (>140 Hz) in control animals (example traces shown in *Figure 1B*, *Figure 1—figure supplement 1A*) and in animals with epilepsy (example traces shown in *Figure 1C*, *Figure 1—figure supplement 1B*). Given that filtering can be problematic for detection

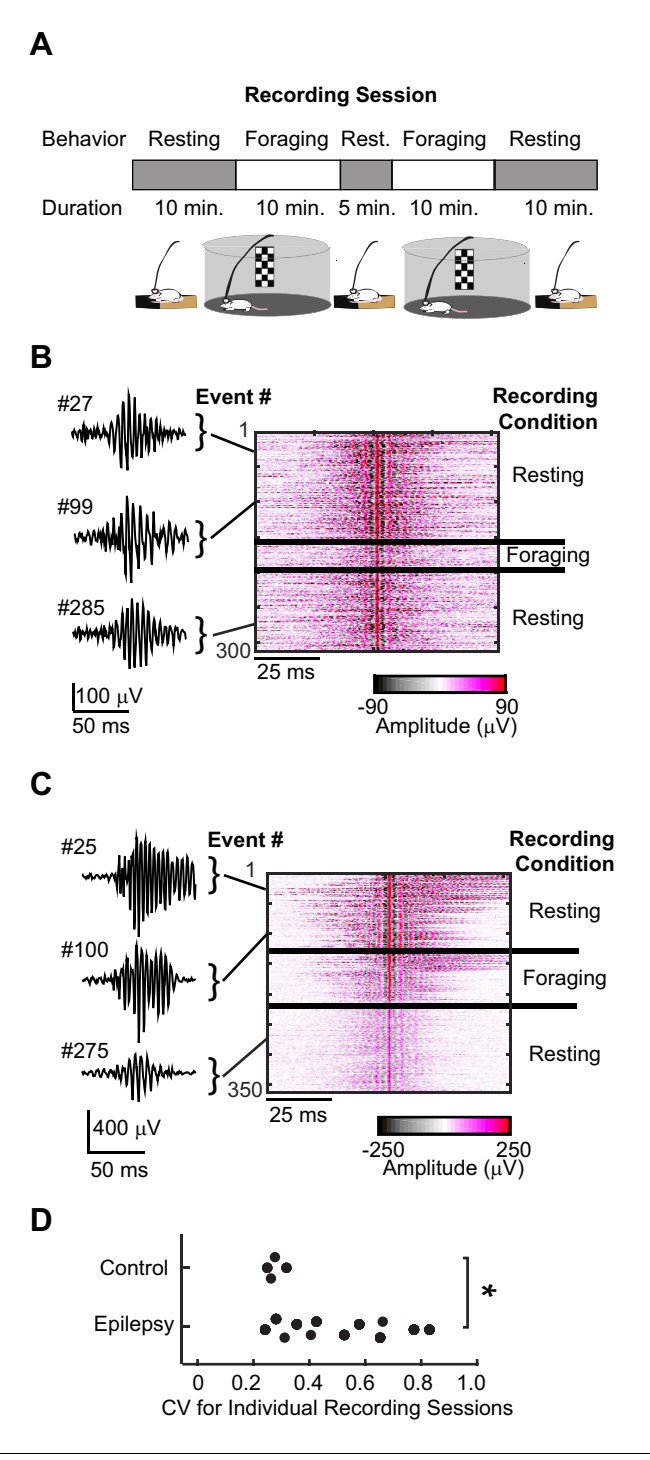

**Figure 1.** High-frequency oscillations in animals with epilepsy compared to control animals. (**A**) Each recording session consisted of a sequence of rest epochs (grey blocks) and foraging epochs (white blocks). Rats were placed in either circular (diameter = 1 meter, pictured) or square (0.8 x 0.8 meters) open arenas for foraging and in a plexiglass holding box (0.2 × 0.2 meters) adjacent to the open arena for rest. (**B**) Example high-frequency oscillations recorded from the CA1 cell layer of a control rat during rest and foraging. Each line of the heat map corresponds to a 100 ms voltage trace of a high-frequency oscillatory event, high-pass filtered, and centered on its peak voltage amplitude. Positive and negative voltages are represented on a color scale (bottom). Three example voltage traces are shown to the left, and see Figure – Supplement1 for more examples. (**C**) The same as (**B**), but recorded from the CA1 cell layer of a rat with chronic epilepsy. (**D**) For each recording session that had at least 20

*Figure 1 continued on next page*

*Figure 1 continued*

high-frequency events (n = 4 control, n = 12 epileptic), the coefficient of variability was calculated for the distribution of peak voltage amplitudes associated with each high-frequency oscillation. *, $p \leq 0.05$, Wilcoxon Rank Sum test.

DOI: https://doi.org/10.7554/eLife.42148.002

The following figure supplements are available for figure 1:

**Figure supplement 1.** Detection criteria for high-frequency oscillations (HFO).

DOI: https://doi.org/10.7554/eLife.42148.003

**Figure supplement 2.** Interictal spikes and noise artifacts are not detected as high frequency oscillations (HFO).

DOI: https://doi.org/10.7554/eLife.42148.004

of ripples in epileptic networks (*Bénar et al., 2010*), we confirmed that our filter did not induce spurious detection of artifact or interictal spikes with no associated high-frequency oscillations (*Figure 1—figure supplement 2*). In control animals, the peak amplitudes of the high-passed ripple oscillations were similar in amplitude across an entire recording session and therefore recording sessions were associated with small coefficients of variation. In contrast, in animals with epilepsy the amplitudes of high-frequency oscillations within recording sessions varied more substantially (ctrl, n = 4; mean ± SEM, 0.3 ± 0.01; epileptic, n = 12; mean ± SEM, 0.5 ± 0.05) (Unpaired t-test, p = 0.04, tstat = −2.2, d.f. = 14) (*Figure 1D*).

The larger variability in high-frequency oscillation amplitude observed in recordings from animals with epilepsy suggested that the pathological CA1 network can engage in different types of fast oscillations, even within the same recording period. It is well known that fast oscillations in epileptic networks can co-occur with interictal spikes (*Jacobs et al., 2008*; *Salami et al., 2014*), leading to the question whether heterogeneity in fast oscillations could be explained by co-occurrence with interictal spikes. In our data, we observed that fast oscillations co-occurred with two distinct types of comparatively slower voltage deflections or 'envelopes'. Some co-occurred with large voltage envelopes that resembled interictal spikes, whereas others co-occurred with smaller voltage deflections that appeared similar to normal 'sharp waves' observed in control animals. We therefore analyzed the relationship between the amplitude of the envelope and the peak frequency of the fast oscillation (*Figure 2—figure supplement 1A–C*). Plotting the relationship between the two shape parameters for each event revealed that events from control animals ('ripples') clustered into one area and events from animals with epilepsy indeed clustered into two areas (*Figure 2A & B*, *Figure 2—figure supplement 1D*), indicating the presence of two types of high-frequency oscillations in the CA1 region in our animal model of epilepsy.

The presence of two clusters of high-frequency oscillations led to the question whether one cluster resembled the normal ripples of control animals. First, k-means clustering was used to segregate high frequency oscillations from the CA1 of epileptic animals into two groups. Interestingly, one group, which we refer to as 'ripple-like', had substantial overlap with control ripples for both peak frequency and envelope amplitude (*Figure 2C & E*) (ctrl ripple n = 1414; peak frequency median, 185.8 Hz; inter-quartile range (IQR), 175.8–197.8 Hz; envelope median, 244.9 μV; IQR, 169.6–380.2 μV; ripple-like n = 1616; peak frequency median, 184.9 Hz; IQR, 168.7–199.1329 Hz; envelope median 253.8 μV; IQR, 148.9–377.2 μV). The other group, which we refer to as 'pathological high-frequency oscillations (pHFOs)', were associated with large amplitude envelopes (interictal spikes) and had minor overlap with control ripples (*Figure 2C & E*) (pHFO n = 923, peak frequency median, 245.9 Hz; IQR, 233.7–264.1 Hz; envelope median, 520.0 μV; IQR, 407.3–647.4 μV). Both groups were present in individual epileptic rats, even in the same recording session (*Figure 2—figure supplement 1D*). A binary classifier was able to distinguish pHFOs from control ripples with high accuracy, as shown by a receiver operator curve (ROC) in the far left quadrant of the ROC plot (*Figure 2D & F*, red line). In contrast, the classifier was unable to distinguish ripple-like from control-ripple, as shown by the curve along the identity line that indicates equal numbers of false positive and true positive classifications (*Figure 2D &F*, blue line). Together, these data show that in animals with epilepsy the CA1 network of the hippocampus engages in two types of fast oscillations, one of which resembles the sharp wave ripples in controls.

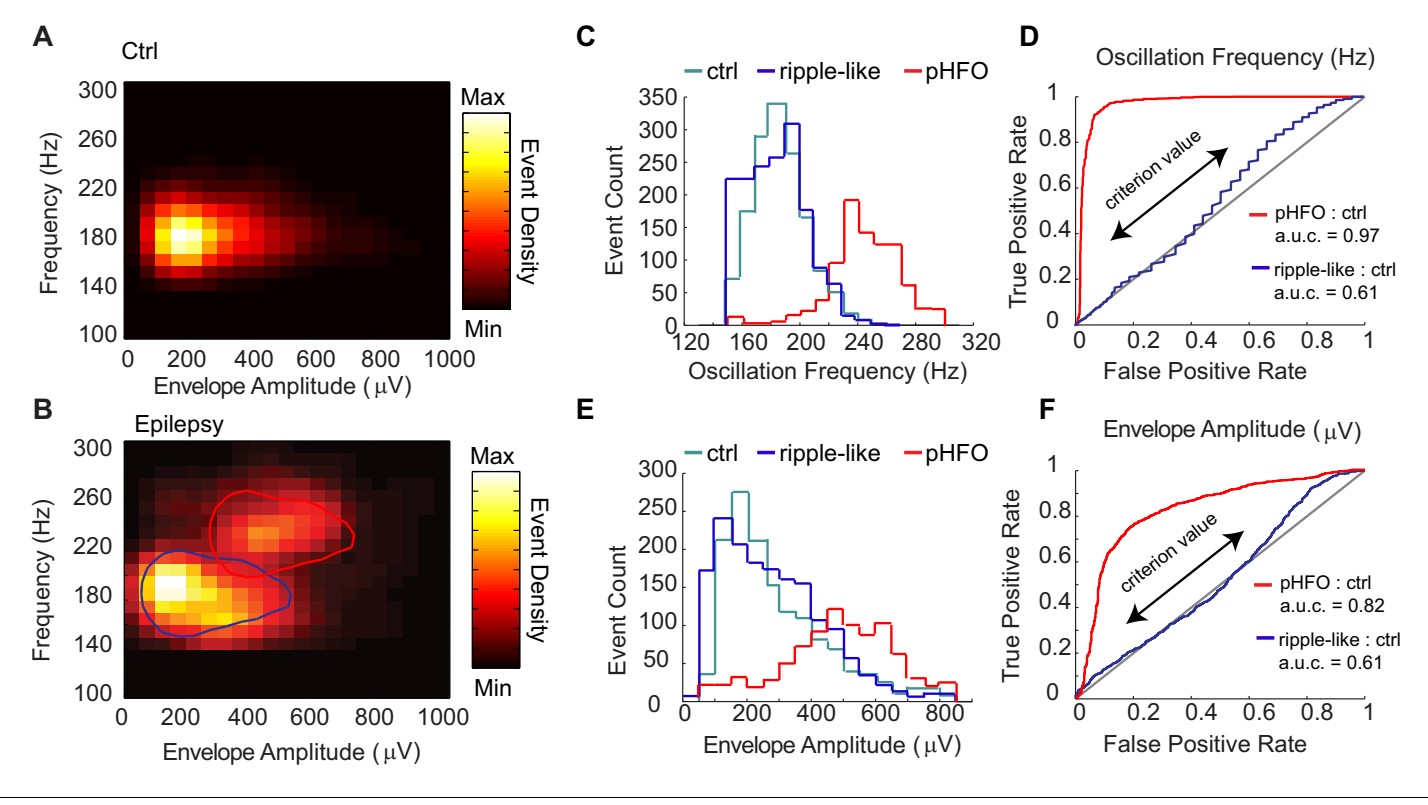

**Figure 2.** Classification of two types of high-frequency oscillations in animals with epilepsy. A Density plot of high-frequency oscillations (HFOs) recorded from all sessions in control rats (n = 4 rats, n = 4 sessions). All events were characterized by a single cluster of similar frequency and envelope amplitude. (B) Density plot of HFOs from all sessions recorded from epileptic rats (n = 4 rats, n = 12 sessions). In animals with epilepsy, two clusters are present, one characterized by envelope amplitudes and frequencies that resemble control ripples (circled in blue), and the other characterized by higher frequencies and larger envelope amplitudes (circled in red), and see Figure – Supplement1 A-C. (C) Distribution of the oscillation frequencies for control ripples (cyan, n = 1414), and in animals with epilepsy (blue and red). The two distributions for animals with epilepsy were separated by clustering the data presented in (B), and see Figure – Supplement1D for display of individual animals. Note that the blue population overlaps with control ripples and is thus referred to as 'ripple-like' (n = 1616). The red population is separate from control ripples and is thus referred to as 'pathological HFOs (pHFO)' (n = 923). (D) The separation (or not) of each cluster in epileptic animals from control ripples is confirmed analytically using a binary classifier and plotted in a Receiver Operator Curve. True positive and false positive rates are plotted with the range of oscillation frequencies used as criteria values (see Materials and methods for more detail). (E, F) The same as (C, D), but for envelope amplitude rather than oscillation frequency.

DOI: https://doi.org/10.7554/eLife.42148.005

The following figure supplement is available for figure 2:

**Figure supplement 1.** pHFO and ripple-like events are both present in each of the chronically epileptic animals.
DOI: https://doi.org/10.7554/eLife.42148.006

## pHFOs are not brain state dependent and occur during foraging epochs characterized by movement-related theta oscillations

To further investigate the extent to which ripple-like events are normal and pHFOs are abnormal we studied their occurrence in relation to ongoing behavior. When animals are mobile, the hippocampal LFP is characterized by sustained theta oscillations that range from 5 to 12 Hz (*Vanderwolf, 1969*). Typically, movement-related theta states preclude ripple oscillations (*Vandecasteele et al., 2014*) such that ripple oscillations are rarely reported to co-occur with theta in healthy animals. Given the direct relationship between movement and theta states and immobility and non-theta slow wave states, we used running speed as an indirect measure of brain state. We analyzed foraging sessions in which the animal's position was tracked allowing calculation of animal running speed at the time of high-frequency oscillations. As expected, ripple events in control animals preferentially occurred during periods of immobility – associated with high slow wave activity and low theta power. Ripple-

like events in epileptic animals retained the same association with immobility as observed in the healthy brain (*Figure 3A*) (ctrl ripples n = 691; median speed, 0.8 cm/s; IQR, 0.6–1.9 cm/s; ripple-like n = 293; median speed, 0.5 cm/s; IQR, 0.2–3.1 cm/s). In contrast, pHFOs occurred over the entire range of running speeds (pHFO n = 490; median speed, 4.7 cm/s; IQR, 1.9–7.4 cm/s)

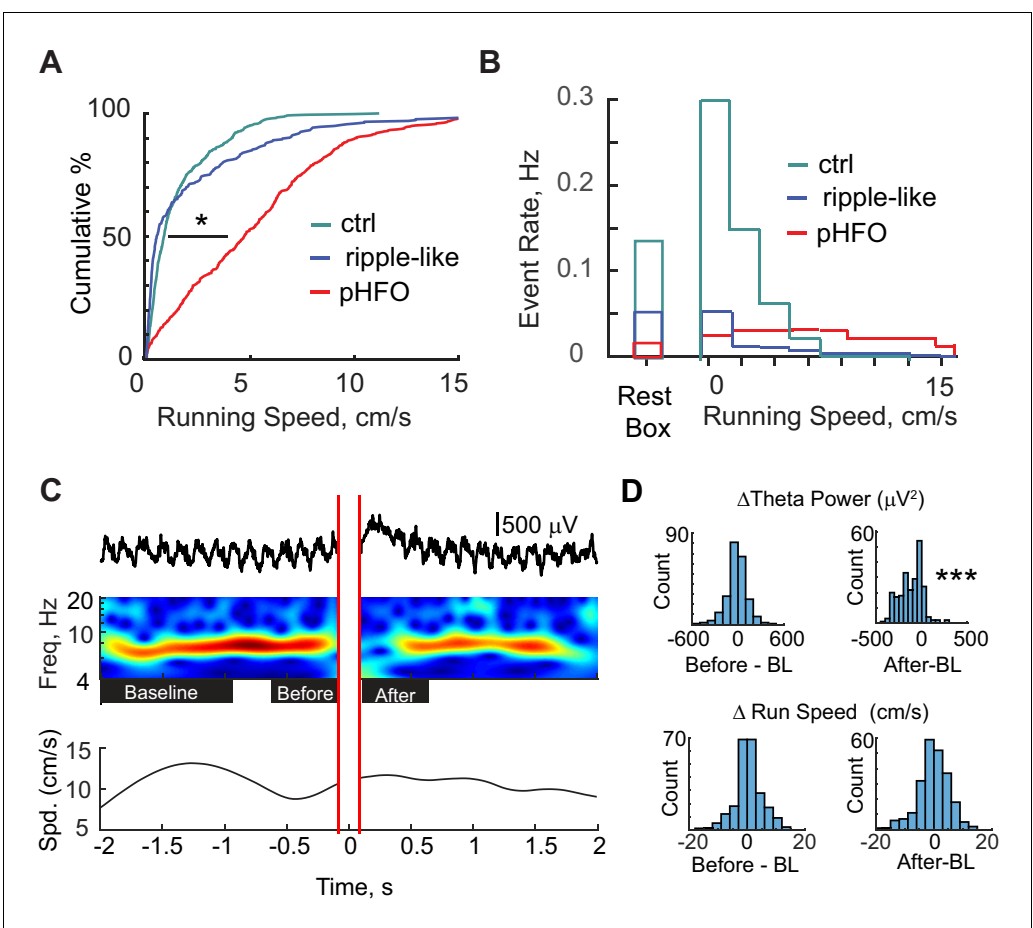

**Figure 3.** pHFOs occurred during movement-associated theta rhythm. (**A**) Cumulative distributions of the running speed at times when a high-frequency oscillation occurred. All data were recorded from behavioral sessions when animals were foraging for food reward and when position and running speed were tracked (n = 691 control ripple, n = 293 ripple-like, n = 490 pHFO). *, p ≤ 0.05, Kruskal-Wallis ANOVA. Control ripple (cyan) and ripple-like (blue) events occur primarily when animals were not moving while pHFOs (red) occurred at all running speeds. (**B**) Event rates for control ripple (cyan), ripple-like (blue), and pHFOs (red) plotted as a function of time spent in the rest box or time spent at each running speed recorded during foraging experiments. Behavior during recordings in the rest box comprised a mixture of mobile and rest behavior. (**C**) Top, LFP recorded in stratum radiatum around the time of a pHFO/interictal spike event. The time frequency plot over the four seconds around the event is characterized by strong theta rhythm leading up to the event and a reduction in theta power after the event. Red parallel lines denote the LFP window that was clipped to remove the large interictal spike before wavelet analysis was performed. Bottom, the running speed of the animal plotted over the corresponding four seconds. See *Figure 3—figure supplement 1* for theta phase analysis. (**D**) LFP and running speed around pHFOs that occurred when animals were running at speeds ≥ 5 cm/s were examined for modulation (n = 231). The distributions of the change in theta power (top) or running speed (bottom) compared to baseline (BL) are plotted for the periods before and after interictal events. Time windows are defined as shown by the black boxes in (**C**). ***, p ≤ 0.001, Wilcoxon Sign Rank test.

DOI: https://doi.org/10.7554/eLife.42148.007

The following figure supplement is available for figure 3:

**Figure supplement 1.** pHFOs occur at distinct theta phases unique for each animal.

DOI: https://doi.org/10.7554/eLife.42148.008

(p = $5.3 \times 10^{-74}$, Chi-sq = 337.4, d.f. = 2, Kruskal-Wallis ANOVA followed by Tukey Kramer multiple comparison test with p < 0.05; control ripple compared to ripple-like, n.s.; ctrl compared to pHFO, p < 0.05). The lack of dependence on immobility of pHFOs was not a consequence of gross differences in behavior between epileptic and control rats, as the two groups ran at similar speeds during foraging periods (ctrl n = 13 10 min foraging periods, median speed 5.6 cm/s; IQR, 5.1–6.9 cm/s; epileptic n = 35 10 min foraging periods, median speed 6.4 cm/s; IQR 4.4–7.6 cm/s, Wilcoxon Rank Sum test; n.s., p = 0.7, z-value = −0.4). Furthermore, when event rates were calculated for the time in the rest box and for different speeds in the arena, high rates of ripple and ripple-like events were observed only during periods of immobility and rest, whereas rates of pHFOs were at approximately the same level during rest as over the entire range of running speeds (*Figure 3B*).

The occurrence of pHFOs during periods of fast running speed was surprising and raised two possibilities. One possibility is that pHFOs occur during movement related theta, without altering ongoing theta oscillations. Alternatively, theta rhythm might be suppressed when pHFOs are generated, which would mean that theta is decoupled from running in animals when pHFOs occur. To distinguish between these two possibilities, a wavelet analysis was performed on LFP recordings during the seconds prior to and following pHFOs that occurred when animals were moving quickly (running speeds of 5 cm/s or faster, n = 231). Even though running speed was consistent around the time of the pHFO (median speed change from baseline to before the pHFO 0 cm/s; IQR −2–3 cm/s; from baseline to after the pHFO, 0 cm/s; IQR −3–3 cm/s), LFP recorded in stratum radiatum was characterized by robust theta oscillations leading up to the pHFO (median power change between baseline and the period immediately preceding the pHFO, 10.9 $\mu V^2$; IQR, −51.5–74.6 $\mu V^2$, n.s., p = 0.08, z-value = 1.6, Wilcoxon signed rank test) followed by a transient suppression of theta oscillations for several hundred ms after the event (median reduction from baseline, 98.0 $\mu V^2$; IQR, 21.3–206.2 $\mu V^2$, p = $4.7 \times 10^{-29}$, z-value = 11.2, Wilcoxon signed rank test) (*Figure 3C & D*). These results are consistent with previous reports that observe theta suppression after interictal spikes in humans (*Fu et al., 2018*). Indeed, in stratum radiatum, the most prominent feature of the interictal event is the interictal spike, consistent with the notion that the interictal spike/pHFO complex also suppresses theta in the rat model. We wondered whether pHFOs preferentially occurred at specific phases of theta, which could have implications of involvement of other brain regions (e.g. entorhinal cortex). Although a phase preference was observed, the exact phase preference differed between animals (*Figure 3—figure supplement 1*). The suppression of theta after each event implies that pHFOs and the associated inter-ictal spikes disrupt ongoing theta oscillations. The transient interruption in theta occurs even while animals continue to explore their environment, and thus could have negative impacts on the temporal coordination of hippocampal ensembles during exploration.

## Sparse activation of CA1 principal cells during high frequency oscillations in animals with epilepsy

Our results thus far indicated that in chronic epilepsy, networks were still capable of generating ripple-like events even though they concurrently generated pHFOs, which suggested that some normal physiological processes could be maintained in these highly pathological networks. In control animals, ripple oscillations in the LFP coincide with activation of ensembles of neurons in CA1 (*Wilson and McNaughton, 1994*). We therefore asked whether ripple-like and pHFO events in animals with epilepsy would show similar participation by principal neurons in CA1 that were recorded over an entire recording session (*Figure 4A & B*, example from an animal with chronic epilepsy and *Figure 4—figure supplement 1* for more examples and cluster quality metrics). In control animals, most principal neurons were ripple-modulated, increasing their firing rate during ripples (32/35, or 91%) (*Figure 4C*), which is consistent with reports from other rat strains (*Csicsvari et al., 1999*). In striking contrast, in animals with chronic epilepsy, a smaller proportion of principal cells were modulated by ripple-like events (59/127, or 46% compared to 91%, p = 2.1 $*10^{-6}$, chi-sq = 22.5, Chi-square test) (*Figure 4D*). Thus, in animals with epilepsy, fewer CA1 neurons were engaged during ripple-like oscillations. One possible explanation for fewer neurons being modulated by ripple-like events is that neurons have been 'hijacked' and now participate in pHFO events instead of ripple-like events. This was not the case, however, as an even sparser activation of neurons was observed during pHFOs, with only 34/153, or 21% participating (*Figure 4E*). Others have reported that there is strong recruitment of inhibitory neurons during interictal spikes (*Muldoon et al., 2015*), which could partially explain the sparser activation of principal cells in pHFO events.

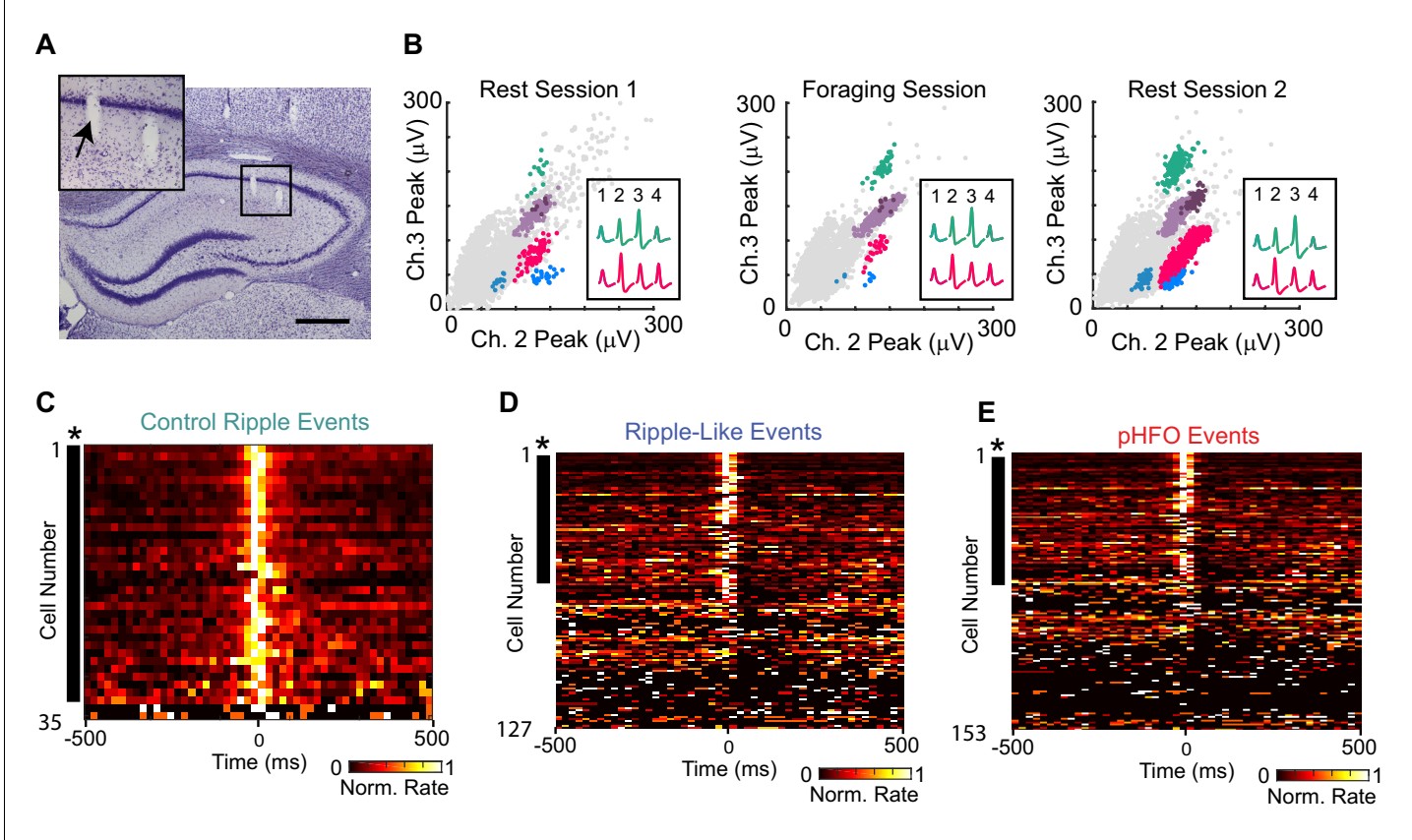

**Figure 4.** CA1 principal neurons increase firing rates during control ripple, ripple-like events, and pHFOs. (A) Coronal section through the area of the hippocampus with tetrode tracks in the CA1 area. The area in the black box is magnified in the inset on the upper left. The tetrode position for the spike recordings in (B) is indicated by an arrow. Scale bar, 300 μm. (B) Scatterplots of spike amplitudes on two of the four recording channels (ch2, ch3) from the tetrode identified in (A). Each colored cluster corresponds to spikes from one cell, and insets show average waveforms of two well-isolated cells. Noise spikes are shown in grey. The similarity of the scatterplots and waveforms across the rest and foraging epochs indicates that the same cells were reliably recorded throughout the session. For more details about single unit sorting, see Figure – Supplement 1.( C) Each row in the heat map is the average rate vector for an individual control CA1 principal neuron aligned to ripple events (time 0). Average rates are normalized to their peak and range from 0 (black) to 1 (white). Neurons are sorted by ripple-modulation significance such that strongly modulated neurons are at the top. Neurons that show significant modulation (p <0.05, *) are marked by a black dash to the left of the row. Almost all control neurons are ripple-modulated (91%), and fire maximally during the ripple period (time 0). (D) The same as (C), but for neurons recorded in animals with epilepsy during ripple-like events (left, 46% of neurons were modulated). (E) The same as (C), but for neurons recorded during pHFO events (right, 21% of neurons were modulated). Please note that neurons in (D) and (E) are ordered differently, therefore cell identity across rows are not comparable. For rates and p-values see Figure – source data (1–3, corresponding to C-E).

DOI: https://doi.org/10.7554/eLife.42148.009

The following source data and figure supplement are available for figure 4:

**Source data 1.** Neuron activity during control ripple.
DOI: https://doi.org/10.7554/eLife.42148.010
**Source data 2.** Neuron activity during ripple-like oscillations.
DOI: https://doi.org/10.7554/eLife.42148.011
**Source data 3.** Neuron activity during pHFO.
DOI: https://doi.org/10.7554/eLife.42148.012
**Figure supplement 1.** Spike waveforms are equally stable in control and epileptic animals and cluster quality is high in both groups.
DOI: https://doi.org/10.7554/eLife.42148.013

In several recording sessions, we were able to record from the same neurons (n = 127) during periods with sufficient numbers of pHFOs and ripple-like events to test whether individual neurons were preferentially modulated by a specific event type. Of the 59 neurons that were modulated by at least one type of high-frequency oscillation during these recording sessions, most were

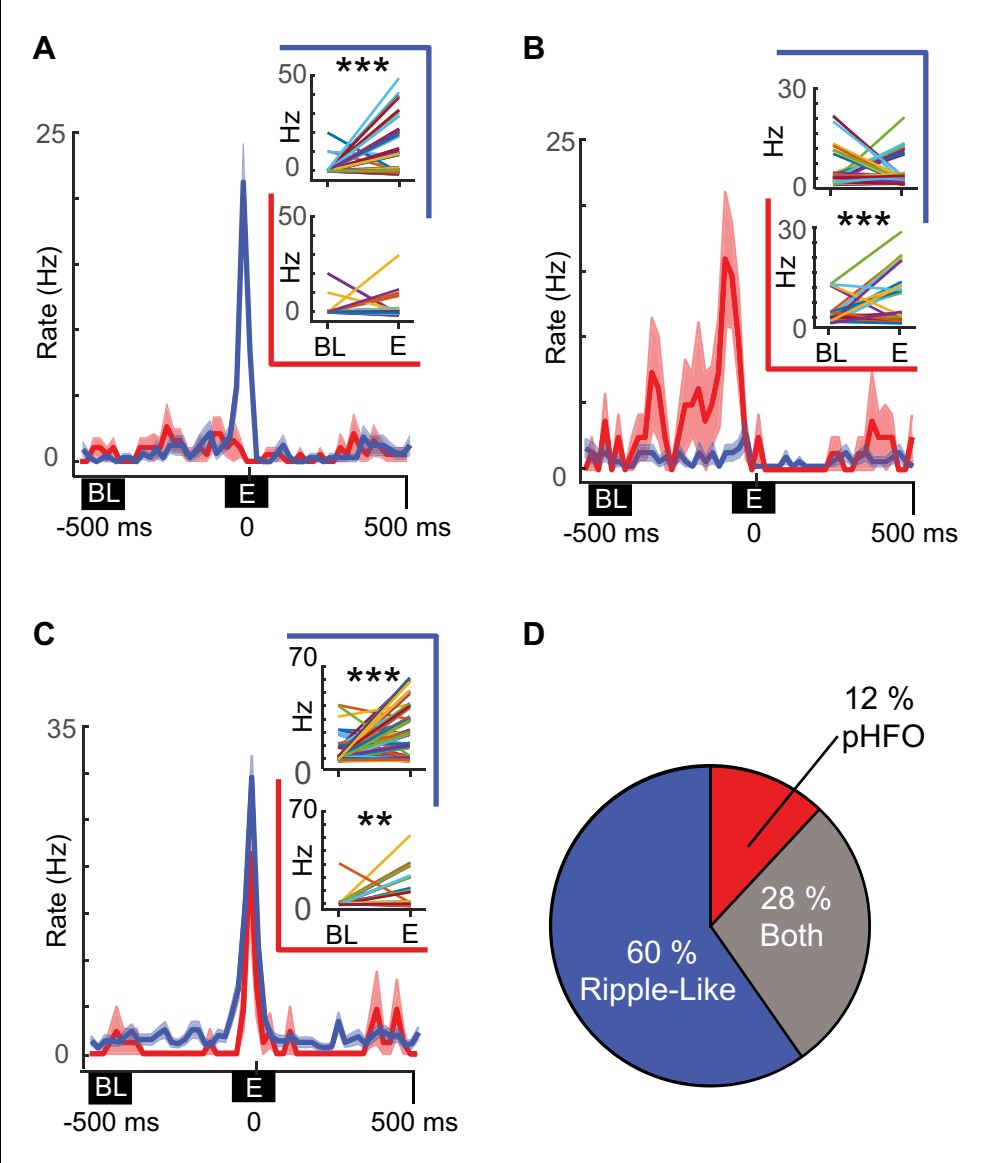

**Figure 5.** Distinct neuronal ensembles are engaged during ripple-like and pHFO events. (**A**) Example CA1 pyramidal neuron that is modulated by ripple-like events only. The mean firing rate ±SEM over all ripple-like (blue) and pHFO events (red) is shown, and traces are centered on the ripple event ('E', time zero). Insets show the mean rate changes for the same neuron during individual events (the rates associated with individual events are depicted with distinct line colors). Rates are shown comparing baseline ('BL') to ripple-like events (upper plot, blue border) or comparing baseline to pHFO events (lower plot, red border). ***, p ≤ 0.001, Wilcoxon Sign Rank test. (**B**) As in (**A**) but an example of a cell modulated by pHFO events only. (**C**) A neuron that was modulated by both ripple-like and pHFO events. (**D**) Population summary of which event types individual neurons were modulated by.
DOI: https://doi.org/10.7554/eLife.42148.014

modulated by only ripple-like events, while smaller subsets were modulated by both or only pHFOs (*Figure 5A–C*). Thus, more than two thirds of modulated neurons were modulated by only one event type (60% by only ripple-like, 12% by only pHFO, *Figure 5D*), suggesting that unrelated cell ensembles in the CA1 network may support the two distinct types of high-frequency oscillations.

## Fewer active neurons and reduced spatial information during spatial coding in animals with epilepsy

Neurons active during behavior typically show a correspondence to those active during ripple oscillations (*Wilson and McNaughton, 1994*). Given our observation that a smaller proportion of neurons were modulated by ripple-like events in animals with epilepsy than during ripples in controls, we wondered whether we would also observe fewer active neurons during foraging epochs in those animals. For our analysis, we considered epochs with good foraging behavior (100% coverage of the spatial arena, 177 CA1 pyramidal neurons in animals with epilepsy, 35 CA1 pyramidal neurons in control animals). Exploration of the environment did not differ between the two groups of animals (mean time spent in each spatial bin ± SEM; Ctrl, 2.1 ± 0.03 s, n = 2560 spatial bins; Epilepsy, 2.2 ± 0.03 s, n = 6656 spatial bins; n.s., p = 0.08, d.f. = 9214, Unpaired t-test). For individual neurons, we plotted the action potential locations as a function of the animal's path during foraging in an open arena (*Figure 6A & B*, left). From these maps of place-modulated activity of individual neurons, we calculated the average rates for each spatial bin in the map and the peak rate (*Figure 6A & B*, right). Consistent with our observation that a smaller proportion of neurons was activated during ripple-like events in animals with epilepsy, we also observed a smaller proportion of 'active' neurons (neurons with peak rates > 2 Hz) during foraging sessions in animals with epilepsy (*Figure 6C*) (73/177 active in animals with epilepsy, compared to 30/35 in control animals, p < 1.5 *$10^{-6}$, chi-sq = 23.1, Chi square test). Further contributing to decreased spatial coding, active neurons in animals with epilepsy (peak rates > 2 Hz) had significantly less spatial information compared to active neurons recorded from control animals (epilepsy n = 73; median spatial information, 0.5 bits; IQR, 0.3–1.0 bits; ctrl n = 30; median spatial information, 1.7 bits; IQR, 1.0–2.5; p = $3.0 \times 10^{-8}$, z-value = 5.5, Wilcoxon rank sum test) (*Figure 6D*), consistent with results reported in other models of epilepsy (*Liu et al., 2003*).

In addition, we also analyzed the spatial firing pattern of neurons active during pHFOs and compared them to those active during ripple-like events. Interestingly, several parameters used to quantify place coding were comparable between neurons modulated by ripple-like versus those modulated by pHFOs (*Figure 6E* – G). Generally, all neurons in animals with epilepsy that were active during foraging had reduced spatial information (*Figure 6E*), reduced stability of place-related activity (*Figure 6F*), and larger more dispersed place-related activity (*Figure 6G*) compared to neurons recorded in control animals) (p = $1.4 \times 10^{-13}$; $4.2 \times 10^{-15}$; $3.9 \times 10^{-12}$; Chi-sq = 59.1, 66.2, 53.0; d.f. = 2; 2; 2 Kruskal-Wallis ANOVA followed by Tukey Kramer multiple comparison test with p < 0.05) (see *Figure 6* – source data for numerical details). Our results thus reveal that functional impairments of hippocampal networks in epilepsy extend beyond the cellular ensembles that participate in pathological activity patterns. Overall, the physiology that underlies place coding is generally perturbed in animals with epilepsy, as fewer neurons encode place relevant information and those that do are less precise and less stable.

## pHFOs during foraging partially disrupt hippocampal place coding

Given our results that pHFOs can be generated while animals are actively running while foraging for food, and that a proportion of CA1 pyramidal neurons increase their firing rates during pHFOs, we asked whether pHFOs during running might contribute to spatial coding deficits. First, we tested whether pHFOs occurred in specific locations, which could be the case if pHFOs were driven by neurons that had place specific activity. For each foraging session with a sufficient number of pHFOs (>15 events), we constructed maps of where pHFOs occurred relative to animal position (*Figure 7A*). From these maps, we calculated the spatial information carried by pFHO events. To determine a 'chance level' spatial information that would account for the number of events, we randomly shuffled when pHFOs occurred (1000 shuffles per session), and calculated the distribution of spatial information values for comparison (*Figure 7B*). In 9 out of 11 sessions, pHFOs occurred randomly throughout the environment, and thus carried the same amount of spatial information as when pHFO times were shuffled (*Figure 7C*). To then test whether ongoing pHFOs would acutely perturb place coding, we first compared the spatial information of individual neuron's maps from foraging epochs when pHFOs occurred to maps that were free of pHFOs (*Figure 7D*). No difference was observed between these two conditions. However, consistent with the results above (*Figure 6*), both distributions from epileptic animals were significantly different than from control animals

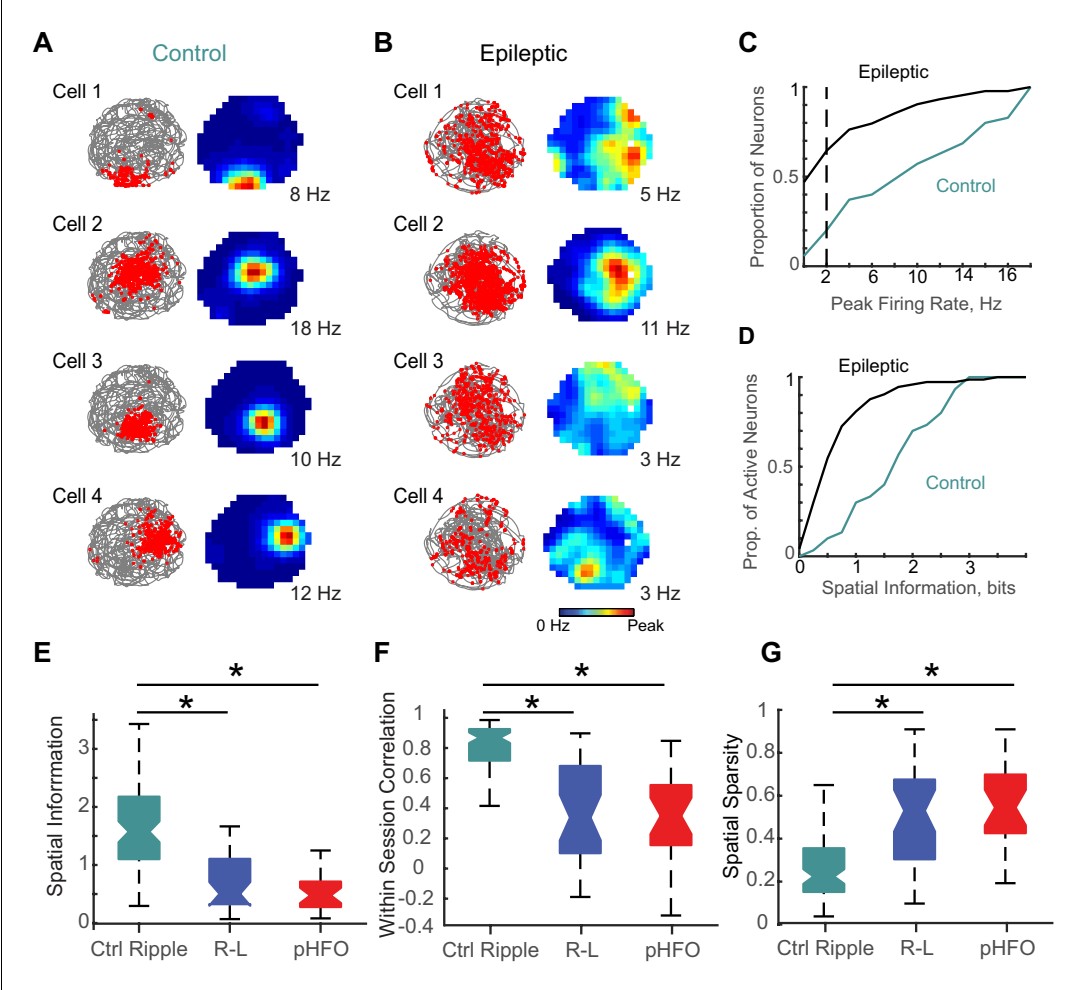

**Figure 6.** Place coding deficits in CA1 neurons from animals with epilepsy. (**A**) The activity patterns of four control CA1 neurons recorded simultaneously during a 10 min foraging epoch in the open field. Left, for each cell the animal's path is in grey and the positions where spikes of an individual neuron occurred are overlaid as red dots. Right, the corresponding firing rate maps with the average firing rate in each spatial location represented from 0 Hz (dark blue) to the peak rate for each cell (red, noted to the right of each map). (**B**) The firing patterns of four CA1 neurons recorded simultaneously from an epileptic rat during a 10-min foraging epoch. Maps are as described in (**A**). (**C**) Distributions of peak rates during foraging for all neurons recorded in control and epileptic rats (for neurons active in several sessions, foraging sessions with highest rates were selected, n = 35 ctrl, n = 177 epileptic). (**D**) Distribution of spatial information for neurons with peak rates $\geq$ 2 Hz (n = 30 ctrl, n = 73 epileptic). (**E,F,G**) Distributions of spatial parameters are shown for three neuron types: (1) CA1 pyramidal neurons recorded from control animals that were ripple modulated 'Ctrl Ripple', (2) CA1 pyramidal neurons recorded from animals with epilepsy that were only modulated by ripple-like events 'R-L', and (3) CA1 pyramidal neurons recorded from animals with epilepsy that were either modulated by pHFOs only or by both pHFOs and ripple-like events 'pHFO'. See *Figure 6* – source data for numerical values. *, $\leq$ 0.05 Kruskal – Wallis test, Turkey-Kramer post hoc.

DOI: https://doi.org/10.7554/eLife.42148.015

The following source data is available for figure 6:

**Source data 1.** Spatial coding parameters of place cells in control and epileptic animals.

DOI: https://doi.org/10.7554/eLife.42148.016

(p = $1.1\times10^{-19}$, chi-sq = 87.4, Kruskal-Wallis ANOVA followed by Tukey Kramer multiple comparison test with p < 0.05) (ctrl n = 93 maps, median spatial information = 1.7 bits; IQR = 1.1–2.2 bits; w/ pHFOs, n = 83 maps med. = 0.5 bits; IQR = 0.3–0.9 bits; pHFO free, n = 59 maps, med. = 0.5 bits; IQR = 0.3–0.9).

The corresponding deficit in place coding for foraging epochs with and without pHFOs is consistent with the result that only a few neurons are engaged by pHFOs and that the firing patterns of most neurons would thus not be directly altered by the occurrence of pHFOs. Thus, we examined

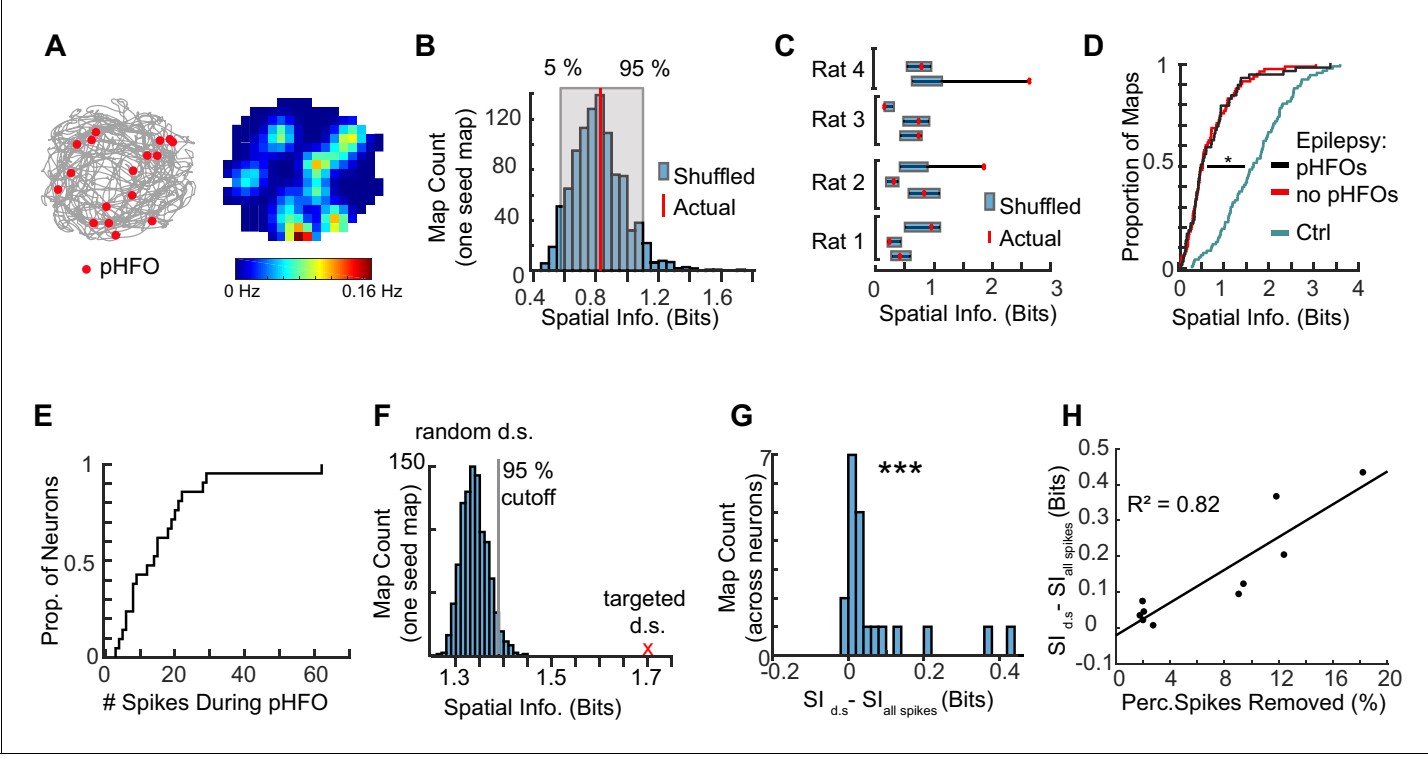

**Figure 7.** The spiking of neurons during pHFOs reduce spatial information. (A) The animal's path is depicted in grey with the positions where pHFO events occurred overlaid as red dots. The corresponding average rate map is displayed to the right, with a peak rate of 0.16 Hz. (B) Distribution of shuffled spatial information for pHFO events shown in (A). The actual spatial information is denoted by the red line. (C) Summary of the relationship between shuffled and actual spatial information of pHFOs for all foraging epochs with frequent pHFOs. Blue bars denote 5 and 95 percentile of shuffled distributions. Red dots show actual information. (D) Distributions of spatial information of neurons recorded from control animals (cyan, n = 93 maps) compared to spatial information of neurons in animals with epilepsy recorded either during foraging epochs with ongoing pHFOs (black, n = 83 maps) or no pHFOs (red, n = 59 maps). * ≤ 0.05 Kruskal – Wallis test, Tukey Kramer post hoc. (E) Distribution of the number of spikes that occurred during pHFOs for individual neurons during foraging epochs with ongoing pHFOs (n = 21). (F) The distribution of spatial information after random down-sampling (random d.s.) was calculated to determine whether targeted removal of spikes that occurred during pHFOs (targeted d.s., shown as a red x) led to a significantly higher spatial information score than would be expected by random down-sampling. (G) Spatial information of individual neurons was improved when spikes that occurred during pHFOs were removed from the map (targeted d.s.). *** ≤ 0.001 Wilcoxon Sign Rank test. (H) The improvement in spatial information as a function of the percentage of spikes which occurred during pHFOs. Values from individual neurons are plotted as black points. Only maps that had significant improvements beyond what would be expected by random down-sampling (as shown in F) are plotted and fit with a line.

DOI: https://doi.org/10.7554/eLife.42148.017

specifically neurons that spiked during pHFOs and asked whether these cells would show reduced spatial tuning. We selected the subset of neurons with at least 1% of spikes occurring during pHFOs (*Figure 7E*) (n = 21, median number of spikes occurring during pHFOs = 14; IQR, 8–20) and examined their place maps. We removed spikes that occurred during pHFOs, and calculated the spatial information after the targeted down-sample. As a control for reducing the number of spikes, we calculated the spatial information when a random set of spikes equal to the number that occurred during pHFOs were removed (random down-sampling was done 1000 times for each map). For 10/21 maps, the improvement in spatial information after removing spikes that occurred during pHFOs was greater than could be expected by random down-sampling (*Figure 7F*). For the remaining 11/21 maps, the improvement in spatial information after removing spikes during pHFOs could be explained by random down-sampling. When considering all 21 maps after removing spikes during pHFO, there was a small but significant increase in spatial information (increase of 0.02; IQR 0.01–0.08; p = $9.2 \times 10^{-5}$, z-value = 3.9, Wilcoxon Sign Rank Test) (*Figure 7G*). Not surprisingly, for the neurons in which targeted down sampling led to an improvement in spatial information, there was a positive relationship between the improvement in spatial information and the percentage of total

pHFO-related spikes that were removed (*Figure 7H*, $R^2 = 0.82$, $p < 1.6 * 10^{-8}$). Therefore, spikes during pHFOs can have a negative effect on place coding in individual neurons, and negative effects are stronger in neurons with a large number of spikes that occur during pHFOs. Collectively, these results indicate that pHFOs, as they occur randomly with respect to relevant stimuli (in this case spatial location) can reduce the information that is conveyed by CA1 place cells.

## Discussion

Our findings highlight that in individual animals with epilepsy, the hippocampus is capable of engaging in two types of high-frequency oscillations: pHFOs and ripple-like events, both of which involve distinct subnetworks of hippocampal neurons. Ripple-like events in epileptic animals are similar to ripples observed in the healthy brain of control animals in that they are characterized by similar oscillation frequencies and slow envelope amplitudes ('sharp waves') as well as by an increased prevalence during non-theta brain states. In contrast, pHFOs are only observed in the epileptic brain, exhibit faster oscillation frequencies than ripples, are associated with interictal spikes, and are not brain state dependent. The preservation of ripple-like events in parallel with the occurrence of pHFOs suggests that memory-related neural computations are at least partially preserved in animals with epilepsy. Therefore, we investigated to what extent ripple-like events corresponded to control ripples with respect to network dynamics during behavior. In the epileptic brain, strikingly, a much smaller proportion of total CA1 neurons were participating in ripple-like events compared to the proportion that are ripple-modulated in controls. The reduction in the number of neurons that engages in network activity during ripples during immobility extends to spatial activity patterns in theta states during movement. During foraging, we also found a smaller proportion of active neurons in animals with epilepsy compared to healthy controls. Across behavioral states, activation of cell assemblies was therefore sparser in the epileptic compared to the control hippocampus. When neurons were active during foraging in epileptic animals, they did have place fields, although their fields were less precise and more unstable. We found that the general decline in hippocampal spatial information was further enhanced by the immediate effects of pHFOs on the network. Immediate effects included a transient reduction in hippocampal theta power, as well as a further reduction in spatial precision for place cells that were activated by the pathological oscillation. Therefore, the proper classification and selective therapeutic targeting of events that result in pHFOs has the potential to normalize hippocampal function in the epileptic brain.

Clinical practices, which use pHFOs to determine ictal-genic areas in the brain, depend on proper classification of pathological high frequency oscillations. We found that combining frequency and slow envelope amplitude (interictal spike in epilepsy, sharp wave in physiological events) of high-frequency oscillations facilitated separation of two distinct event types in animals with epilepsy. Similar classification approaches have been successfully implemented in human epilepsy patients (*Nonoda et al., 2016*), and others have proposed that the association between high-frequency oscillations and concurrent interictal spikes is indicative that the oscillations are pathological (*Schevon et al., 2009*). However, we are the first to show that such waveform-based classification could separate events recorded at the same electrode. Interestingly, we found that pHFOs occurred during periods of active exploration characterized by robust theta oscillations. We therefore propose that brain state dependence could be added as an additional criterion for classification of pathological oscillations in hippocampus in epilepsy. Contrary to our result, there are reports that fast oscillations in epilepsy are more rare during awake and REM sleep states and instead occur predominantly during non-REM sleep (*Bagshaw et al., 2009*; *Frauscher et al., 2015*). However, it is important to consider that if physiological ripples were not properly separated and removed from the analysis; fast oscillations would appear to be most frequent during non-REM sleep compared to other brain states because of the prevalence of ripple-like events during this brain state (*Figure 3*). Consistent with our results of the occurrence of pHFOs irrespective of brain state, one study in human patients found that pHFOs can occur during theta states (REM sleep), but only when recording in the ictal onset zone (*Sakuraba et al., 2016*). Therefore, it is possible that only pHFOs occur independent of brain state in ictal-genic networks, underscoring the potential for using brain state dependence to improve classification.

It was previously unclear whether pathological and physiological ripple events could coincide in the same seizure-genic network. Modeling has suggested that the same network of neurons would

be able to flexibly transition between activity modes that are either lower frequency and therefore normal and higher frequency and therefore pathological (*Fink et al., 2015*). Here, we confirmed that both pHFO and ripple-like events can co-occur in memory circuits. In support of computational models, we observed a small number of CA1 principal cells that participated in both event types (28%). However, the activity of the majority of CA1 neurons were modulated by only one event type or the other. This segregation suggests that tracking single-unit activity could improve separation of oscillation types in epilepsy, but more importantly motivates future experiments aimed at more selective therapeutic interventions designed to target pathological sub-networks in the hippocampus.

Tracking large numbers of single-units across pHFOs and ripple-like events in behaving animals not only allowed for the identification of additional criteria to better classify pathological and physiological high-frequency activity, but also provided more precise insights into the general and immediate effects pathological oscillations have on hippocampal circuits. The idea that interictal spikes (and co-occurring pHFOs) would disrupt memory processing is widespread (*Holmes and Lenck-Santini, 2006*). Most studies have found that interictal events disrupt memory consolidation processes, as interictal spikes have negative impacts when they occur during retrieval versus encoding phases of memory tasks (*Kleen et al., 2010*; *Kleen et al., 2013*). Furthermore, during sleep, interictal spikes inappropriately initiate spindle activity in cortex, and therefore disrupt memory consolidation processes that engage cortical areas (*Gelinas et al., 2016*). Moreover, a recent study uncovered a cellular mechanism of the perturbed dynamics during CA1 high-frequency oscillations in epilepsy, involving improper inhibition resulting in non-specific recruitment of CA1 neurons during fast ripples (*Valero et al., 2017*). In their study they exclude sessions that had ongoing interictal spikes, whereas in our study, we define pHFOs as those associated with large amplitude envelopes (i.e. interictal spikes), and it is unclear whether the cellular dynamics they report during fast ripple would be similar to those during interictal spike associated pHFO. A better comparison between our two data sets would be made between our ripple-like events and their fast ripples, although in our hands ripple-like events were spectrally more similar to control than their fast-ripples, perhaps because of differences in time-points of recordings. Importantly, we report fewer CA1 neurons are active during ripple-like events compared to control, so in both studies high-frequency oscillations that are not associated with interictal spikes have abnormal dynamics, which could negatively influence memory consolidation.

Given that most studies of interictal activity focus on the effect on consolidation processes, which occur during sleep, our finding that pHFOs occur during active exploration are provocative when considering the effects on memory encoding. We are the first to record from large cell populations during pHFOs while animals are freely exploring, which allowed us to ascertain whether networks generating awake-pHFOs would have specific deficits with regard to place coding. In line with previous reports, we confirmed that place coding is generally disrupted in animals with temporal lobe epilepsy (*Lenck-Santini and Holmes, 2008*; *Liu et al., 2003*). In addition to this general change in place field size, we observed that the place code was generally more 'sparse', both during encoding and during consolidation periods. During foraging epochs (encoding), fewer neurons had place fields, and in subsequent resting periods (consolidation), fewer neurons were active during ripple-like oscillations compared to control animals. A reduction in the place-related activity of CA1 neurons in a rat model of epilepsy has been reported previously (*Liu et al., 2003*), but we are the first to observe that place field networks are also more sparsely activated during behavior as well as during ripple-like events in rest. The reduced number of neurons with place-related as well as memory-related activity may exacerbate the consequence of the reduced spatial precision of the remaining active neurons. It is widely believed that neural networks can overcome high variability in individual neurons by implementing population coding schemes (*Abbott and Dayan, 1999*). Conversely, as individual neurons become more variable, the size of the population required for optimal decoding increases (*Yarrow and Seriès, 2015*). For place coding in temporal lobe epilepsy – individual place cells become more variable, and the population size encoding animal position is reduced, which would collectively deteriorate spatial information coding in cell assembles. Moreover, the reduced number of neurons participating in ripple events would also worsen the efficacy of memory consolidation and readout. Therefore, the association of pHFOs with a higher network sparsity and a decrease in information coding serves as potential mechanisms for memory impairment in epilepsy.

Our findings also suggest that pHFOs have an immediate impact on an already compromised hippocampal network. Despite most studies focusing on interictal events during consolidation, a few

studies have shown that interictal spikes can impact perception at the moment that the spikes occur (*Shewmon and Erwin, 1988a*; *Shewmon and Erwin, 1988b*; *Shewmon and Erwin, 1988c*; *Shewmon and Erwin, 1989*). Our data support the possibility that pHFOs associated with interictal spikes occurring during wakefulness could also confer immediate negative effects on the place code. We observed that pHFOs occur at random locations in environments and modulate a subset of cells that should otherwise not be active at those locations. Such aberrant recruitment could influence the formation of neuronal sequences that are established during exploration and thus compromise memory-guided behavior and navigation. A potential further disruption in neural computations could also be related to the observation that pHFOs have an immediate effect on hippocampal theta power. Theta timing and the organization of hippocampal phase precession has been shown to be generally impaired in temporal lobe epilepsy (*Lenck-Santini and Holmes, 2008*). We show that unlike ripple-like oscillations, pHFOs occur during theta states in behavior when hippocampal sequences are established. In immediate response to the pHFO, we observed a sustained suppression in hippocampal theta despite a lack of change in ongoing behavior and running speed. During this period of time, spatial navigation would not be accompanied by the standard spatial encoding schemes that have been described in the CA1 network during spatial exploration, which are dependent on the local theta rhythm. A lack of hippocampal theta has previously been shown to be associated with a severe reduction in hippocampal-dependent memory and pathfinding (*Chrobak et al., 1989*; *Mizumori et al., 1989*; *Wang et al., 2015*). It should be noted, however, that our data suggests that immediate effects of pHFO are relatively subtle – few neurons are active during them and when they occur during exploration, theta suppression is short lived. Therefore, longitudinal studies to address the long-term effects of pHFO activity on hippocampal network activity are essential to motivate whether early stage intervention of pHFO activity could have a positive impact on preserving healthy levels of network sparsity and spatial coding. For example, such studies would reveal whether neuronal activation during pHFO becomes sparse over time as part of a compensatory mechanism. Furthermore, it remains to be explored how the acute effects of pathological activity differs with distance from the epicenter of epileptic activity, a problem better addressed by simultaneous recordings of cell ensembles across multiple brain regions including hippocampus.

Given the negative effects of epilepsy on cognition, it is of interest that ripple-like oscillations persist in epileptic networks even though they are associated with a smaller proportion of participating cells, suggesting remnants of a healthy network. Furthermore, ripple-like oscillations engage unrelated neurons from those active during pHFOs, which suggests that the event types are supported by distinct network mechanisms. Future research is needed to determine whether neurons participating in the two event types correspond to different classes of CA1 pyramidal cells, as there is heterogeneity within CA1 and certain classes are more likely to be active during normal ripples (*Cembrowski et al., 2016*; *Ciocchi et al., 2015*; *Dong et al., 2009*; *Lee et al., 2014*; *Valero et al., 2015*). Nonetheless, due to the non-overlapping populations of CA1 cells recruited by each event in the same network, the development of selective manipulations that target specific types of pathological activity are possible. Such experiments could directly test the causal role of observed changes in network function for memory comorbidities in epilepsy and have the potential to lead to a reduction in seizures along with a recovery in memory.

# Materials and methods

## Key resources table

| Reagent type (species) or resource | Designation | Source or reference | Identifiers |
| --- | --- | --- | --- |
| Strain, strain background | Wistar rats; Crl:WI | Charles River Labs | Strain Code: 003; RRID: RGD_13508588 |
| Chemical compound, drug | Kainic acid; kainate | Tocris | Cat # 0222 |

*Continued on next page*

*Continued*

| Reagent type (species) or resource | Designation | Source or reference | Identifiers |
|---|---|---|---|
| Chemical compound, drug | Isoflurane | MWI | Cat #: NDC 13985-528-60 |
| Chemical compound, drug | Buprenorphine | MWI | Cat #: 29308 |
| Chemical compound, drug | Chloroplatinic acid for platinum plating | Sigma-Aldrich | Cat #: 206083; CAS 18497-13-7 |
| Chemical compound, drug | Sodium pentobarital | MWI | Cat #: 15199 |
| Chemical compound, drug | Formaldehyde | EMD | Cat #: FX-0415–4; CAS 50-00-0 |
| Chemical compound, drug | Cresyl violet | EMD | Cat #: M-19012; CAS 10510-54-0 |
| Software, algorithm | Matlab v2015b | Mathworks | MATLAB, RRID:SCR_001622 |
| Software, algorithm | MClust | AD Redish | http://redishlab.neuroscience.umn.edu/MClust/MClust.html |
| Software, algorithm | Chronux | Partha Mitra | http://chronux.org/ |
| Other | Hyperdrive | Custom built; Designed by B McNaughton | US Patent: US5928143 A |
| Other | Platinum- Iridium tetrode wire | California fine wire company | Cat #: CFW0011873 |
| Other | Freezing microtome | Leica | Model: SM 2000R |
| Other | Digital Neuralynx recording system | Neuralynx | Model: Digital Lynx SX |

## Subjects

All experimental procedures were performed as approved by the Institutional Animal Care and Use Committee at the University of California, San Diego and according to National Institutes of Health and institutional guidelines. Epilepsy was induced in male Wistar rats (40 days of age, Charles River, CA) using a repeated low-dose kainate chronic model of temporal lobe epilepsy (*Hellier et al., 1998*). Rats were treated with kainic acid (5 mg/kg, i.p., Tocris) each hour until the onset of status epilepticus, which was defined as >10 motor seizures per hour of class IV or V on the Racine scale (*Racine, 1972*). After the seizure induction protocol, the rats were housed individually and maintained on a reverse 12 hr light/12 hr dark cycle. To confirm whether rats had developed chronic spontaneous seizures that define epilepsy, the animals were video-monitored for 4 hr per day beginning at 2 months after kainic acid treatment. Once two or more motor seizures of class III or greater on the Racine scale were observed, the animal was considered epileptic. Four epileptic rats (6–12 months old) and four control rats that were not injected with kainate (6–12 months old) were used for electrophysiological experiments. Four animals per condition was chosen to match the standard of the field of in vivo electrophysiology. Despite obtaining large n from individual animals

(corresponding to various electrophysiological measures), it is important to record from several animals in each condition to account for between animal variability.

## Surgery and electrode placement

Rats were anesthetized with isoflurane (2% to 2.5% in $O_2$) and an electrode assembly that consisted of 14 independently movable tetrodes was implanted above the right hippocampus (AP, 4.1 mm posterior to bregma; ML, 3.0 mm) and fixed to the skull using stainless steel screws and dental cement. Two screws were used as animal ground and were implanted to touch the surface of cortex, anterior and lateral to bregma. Tetrodes were prepared by twisting four insulated platinum wires together (diameter = 0.017 mm, California Fine Wire Company). Leads were plated with platinum prior to surgery to obtain stable impedances near 200 MΩ. Two tetrodes had all four leads shorted with one of the two left in the cortex to record a differential reference signal, while the other was advanced to stratum radiatum to record local field potentials (LFP). The other 12 tetrodes were positioned in the CA1 cell layer to record single units.

## Cell sorting and cell-tracking

Single units were manually sorted using MClust (MClust 3.5, written by A. David Redish; http://redishlab.neuroscience.umn.edu/MClust/MClust.html) adapted to aid in the tracking of cell identity over extended periods of time (*Mankin et al., 2012*). Clusters that persisted in the same region of parameter space throughout a behavioral session were accepted for analysis, such that the action potentials of a given neuron included in the analysis did not change shape, were distinct from noise signals, and remained inside the cluster boundary. The cluster was required to be clearly defined in both the first and last rest period of the recording session, but was not required to be present in individual foraging epochs, as many CA1 single units are silent within a particular environment. Units were not tracked across days. In cases in which several sessions were recorded for individual animals, units were only analyzed if the tetrode did not have units on previous recording days – in other words, units in this study are not double counted. In control animals (n = 4), for each animal, units were isolated in one recording session. In epileptic animals (n = 4), units were isolated in three, five, two, and three recording sessions.

## Recordings and data acquisition

Single units and LFP were recorded during restful and running behavior. All recording sessions occurred during the animal's dark cycle (between 7 am – 7 pm). Animals were food deprived to 85% of their baseline weight and trained to forage for randomly scattered food reward (chocolate cereal crumbs) in open arenas. Open arenas were enclosed by either circular walls (diameter = 1.0 m, height = 0.5 m) or square walls (0.8 × 0.8 m, height = 0.5 m). Behavioral sessions consisted of two to four 10 min foraging epochs preceded and followed by resting periods. The rats were also allowed to rest in a holding box for 5 min between foraging epochs. For position tracking, light emitting diodes on the head-mounted preamplifier were tracked at 30 Hz by processing video images.

## Detection of high-frequency oscillations

Analysis was performed on LFP recordings from the CA1 cell layer in control and chronically epileptic rats. Recordings were band-pass filtered between 140 and 800 Hz, and the root mean square (RMS) was calculated using a five point (2.5 ms) sliding window. The filter used was an Equiripple Bandpass Filter designed using the FIRPM function in the Signal Processing Toolbox of Matlab. The filter was designed to have: first stopband = 0–140 Hz, amplitude 0; passband = 150–800 Hz, amplitude 1; second stopband = 810 Hz – Nyquist Frequency, amplitude 0. The first and second stopband attenuation were set to $5.6 \times 10^{-4}$. The passband attenuation was set to $2.8 \times 10^{-2}$. These led to error minimization weights of 51.2 and 1, respectively. Putative high-frequency events were detected as periods when the RMS signal deviated from the mean RMS by greater than 3.5 x the standard deviation of the RMS. Putative high frequency events within 6 ms of each other were merged. Events were required to have at least five peaks (cycles of oscillation) that each had amplitudes greater than 3 x the standard deviation of the rectified band-passed LFP signal. After initial detection, 20 randomly selected events from each recording session were visually inspected by the experimenter to

determine whether detected high-frequency events were not artifacts (also see *Figure 1—figure supplement 2* for further validation of detection). Furthermore, to confirm that the high-frequency events were not misidentified fast gamma rhythm or noise artifact, the FFT for each event was calculated, and only events for which the peak power for frequencies greater than 150 Hz were larger than the peak power from 75 to 125 Hz were included for further analysis (see *Figure 1—figure supplement 1*).

## Classification of high-frequency oscillations in animals with chronic epilepsy

Peak slow wave amplitude and peak frequency were calculated for each event. To calculate the slow wave amplitude, 500 ms segments of data centered on the high-frequency event were band-pass filtered between 0.2 Hz and 40 Hz, and the maximum absolute amplitude of the filtered data was taken. To calculate the peak frequency, the FFT of the data segment was calculated over the frequency range of 40 Hz to 600 Hz. The peak in power for frequencies greater than 150 Hz was taken as the peak frequency. Peak slow wave amplitude and peak frequency were each normalized to the maximum across the entire data set, and the resulting data were clustered into two populations using a k-means algorithm. For clustering, initial starting centroids were given by user input based on by-eye estimation of the center of masses (highest density of points) in the feature plot. Clustered distributions were then compared to the distribution recorded from control animals using ROC analysis.

## ROC discrimination

To compare the degree of similarity of oscillation shape parameters between events recorded in animals with epilepsy compared to events recorded in control animals we performed a discrimination using receiver operating characteristics analysis (ROC). Comparisons were made between (1) maximum frequency of events in epilepsy cluster one compared to events in control, (2) maximum frequency of events in epilepsy cluster two compared to events in control, (3) envelope amplitude of events in epilepsy cluster one compared to events in control, and (4) envelope amplitude of events in epilepsy cluster two compared to events in control. The true positive rate (ranging from 0 to 1; rate of correct classification of data from distribution A as belonging to distribution A) and false positive rate (ranging from 0 to 1; rate of misclassification of data from distribution B as belonging to distribution A) is plotted for ranges of criteria thresholds (across values of (1) frequency or (2) envelope amplitude). As an example, imagine sliding a vertical line (criterion threshold) from left to right along the x axis of the histograms of the two distributions being compared (see *Figure 2C and E*). For each x position of the line, consider the probability of data to the right of the line to come from one distribution (true positive rate) as opposed to a second distribution. When distributions have no overlap, all thresholds between the two distributions would yield true positive rates equal to one. Distributions with complete overlap would yield true positive rates equal to 0.5 for all criteria thresholds because it would be equally likely for data to belong to either distribution along the entire range of x. For each ROC, the area under the curve (a.u.c.) was used to evaluate discrimination quality – values greater than 0.8 are considered as acceptable discrimination. Analyses were performed in Matlab (2015b) using the built in functions 'patternnet' and 'roc'.

## Analysis of theta oscillations

For each detected pHFO that occurred when animals were running at least 5 cm/s or faster, four seconds of LFP signal (recorded in stratum radiatum) that were centered on the pHFO event were taken. The large voltage spike (the interictal spike) associated with the pHFO was removed by setting the 200 ms at the center of the signal to the mean of the entire 10 min LFP recording. The spike was removed to avoid distortion of time frequency plots by the large voltage change associated with the interictal spike. Time frequency plots of the four seconds were generated using Morlet wavelets over a frequency range of 4–20 Hz. Mean power over the frequency range of 7–11 Hz was calculated for three time windows: 'baseline' defined as the 1 s between 2.0 and 1.0 s before the pHFO, 'before' defined as the 500 ms between 0.6 and 0.1 s immediately before the pHFO, and 'after' defined as the 500 ms between 0.1 and 0.6 s immediately after the pHFO. To determine the phase that pHFO occurred, the two seconds of LFP data (recorded from stratum radiatum) previous

to the time of the pHFO event were taken for further analysis. A threshold was determined by multiplying the standard deviation over the first second (std) by 5, and only interictal spikes that exceeded the threshold were included in analyses (excluded 45 events of the 231). To determine the phase of theta rhythm when the pHFO/interictal spike was generated, the onset of the interictal spike (envelope) was determined with user input to make sure that the calculated onset was not early (which would cut out partial theta cycles) or late (which would add large voltage deflections at the end of the signal and distort filtering). The signal was truncated at the onset of the interictal spike; therefore, the theta phase associated with the last point in the truncated segment was our estimation of the theta phase at the time of pHFO initiation. To determine the phase at the end of the signal we determined the theta phase of the whole signal by a Hilbert transform on the band-pass filtered LFP using both a zero-phase and a causal butterworth filter (4–12 Hz). The causal filter can be applied until the end of the signal just before the pHFO, because it only uses information from past sampling points, but provides incorrect phase estimates that include phase distortions introduced by the asymmetry of the filter kernel. Conversely, the symmetric zero-phase filter provides correct phases but cannot be applied until the end of the signal because it includes samples perturbed by the pHFO. We used the unperturbed time interval to estimate the phase difference between causal and zero-phase filter and used this phase offset to linearly extrapolate the theta phase at the pHFO over the last six cycles from the causally filtered trace.

## Neuron modulation by high-frequency oscillations

Only neurons that were recorded during a period in which at least 10 high-frequency oscillations occurred were included in the analysis of neuron modulation by HFO. Spike times that fell within 1 s of each high-frequency oscillation (500 ms before, 500 ms after) were binned into 20 ms bins. The number of spikes that occurred within each bin was divided by 20 ms to calculate a binned firing rate vector. For statistical analysis, the binned firing rate vector associated with each high-frequency oscillation was treated as an independent observation. Paired comparisons between baseline rates (mean from 500 ms to 400 ms before HFO) and 'during HFO rates' (mean of 100 ms centered on the HFO) were done with a Wilcoxon Signed Rank Test, and neurons that had p-values less than 0.05 were considered to be significantly modulated by the high-frequency event.

## Neuronal firing rate map

For each neuron, we constructed firing rate maps for each behavioral session by summing the total number of spikes that occurred in a given location bin (5 × 5 cm), dividing by the total amount of time that the rat occupied the bin, and smoothing with a 5 × 5 bin Gaussian filter with a standard deviation of approximately one bin:

[0.0025 0.0125 0.0200 0.0125 0.0025;
0.0125 0.0625 0.1000 0.0625 0.0125;
0.0200 0.1000 0.1600 0.1000 0.0200;
0.0125 0.0625 0.1000 0.0625 0.0125;
0.0025 0.0125 0.0200 0.0125 0.0025].

Bins that were never within a distance of less than 2.5 cm from the tracked path or with total occupancy of less than 150 ms were regarded as unvisited and were not included in the rate map.

## Proportion of active neurons

The total number of single units was determined as the number that were tracked across resting and foraging behavior. All included single units were active during the first and last resting period (validating their stability), but not all single units had high firing rates during foraging epochs. Thus, we determine the proportion of active neurons as the number of neurons that had at least one neuronal firing rate map with a peak >2 Hz divided by the total number of single units recorded.

## Spatial information score

Spatial information was calculated for each neuron for each foraging session in which the neuron had a peak firing rate of at least 2 Hz in at least one spatial bin. We calculated the spatial information per spike for each firing rate map as: $I = \sum_i P_i \frac{R_i}{R} log_2 \frac{R_i}{R}$, where $i$ indexes the spatial bins, $P_i$ is

the probability of occupancy in each bin, $R_i$ is the mean firing rate in each bin, and $R$ is the mean firing rate across the spatial map (*Skaggs et al., 1993*).

## Within session spatial correlation

To estimate consistency of spatial firing, a within session spatial correlation was calculated for each neuron for each foraging session in which the neuron had a peak firing rate of at least 2 Hz in at least one spatial bin. The foraging session was split in half and rate maps were calculated for each half of the recording (~5 min). We calculated the spatial correlation between the two rate maps using Pearson's correlation between the firing rates of bins at corresponding locations. Any bins that were unvisited in either map were excluded from the calculation.

## Spatial sparsity

Spatial sparsity was calculated for each neuron for each foraging session in which the neuron had a peak firing rate of at least 2 Hz in at least one spatial bin. We calculated the spatial sparsity as $\left(\sum_i P_i R_i\right)^2 / \sum_i P_i R_i^2$, where $i$ indexes the spatial bins, $P_i$ is the probability of occupancy in each bin, $R_i$ is the mean firing rate in each bin.

## Statistics

For each statistical comparison, normality was assessed with Lilliefors test and either parametric or non-parametric tests were chosen accordingly. Multiple comparisons were made with Tukey Kramer post-hoc tests. Throughout the text values are presented as median and inter-quartile interval unless otherwise noted.

## Data and software availability

MClust software is freely available from AD Redish at: http://redishlab.neuroscience.umn.edu/MClust/MClust.html. Chronux software is freely available at: http://chronux.org/. The processed data (single unit and LFP) and HFO detection analysis code that support the findings of this study are available on Dryad Digital Repository: https://doi.org/10.5061/dryad.c89124p.

# Acknowledgements

We thank B Boublil, M Wong, and A-L Schlenner for technical assistance. Research was supported by grants from the National Institute of Health (MH100349, NS084324, NS086947), the Epilepsy Foundation grant 157927, the Hellman Family Foundation, a Walter F Heiligenberg Professorship to JKL, and National Institute of Health grant NRSA-MH096526 to LAE. The authors declare no competing financial interests.

# Additional information

### Funding

| Funder | Grant reference number | Author |
|---|---|---|
| National Institute of Mental Health | 100349 | Jill K Leutgeb |
| National Institute of Neurological Disorders and Stroke | 084324 | Stefan Leutgeb |
| Epilepsy Foundation | 157927 | Jill K Leutgeb |
| Hellman Foundation | Faculty Award | Jill K Leutgeb |
| National Institute of Neurological Disorders and Stroke | 086947 | Stefan Leutgeb |
| National Institute of Mental Health | NRSA-MH096526 | Laura A Ewell |

The funders had no role in study design, data collection and interpretation, or the decision to submit the work for publication.

## Author contributions
Laura A Ewell, Conceptualization, Data curation, Formal analysis, Funding acquisition, Investigation, Methodology, Writing—original draft, Writing—review and editing; Kyle B Fischer, Data curation, Formal analysis, Methodology; Christian Leibold, Formal analysis, Expertise and feedback; Stefan Leutgeb, Resources, Funding acquisition, Writing—review and editing, Provided expertise and feedback; Jill K Leutgeb, Conceptualization, Resources, Supervision, Funding acquisition, Investigation, Methodology, Writing—original draft, Project administration, Writing—review and editing, Expertise and feedback

## Author ORCIDs
Laura A Ewell (iD) https://orcid.org/0000-0002-1638-426X
Christian Leibold (iD) https://orcid.org/0000-0002-4859-8000
Stefan Leutgeb (iD) http://orcid.org/0000-0003-3367-6536
Jill K Leutgeb (iD) http://orcid.org/0000-0002-2014-842X

## Ethics
Animal experimentation: All experimental procedures were performed as approved by the Institutional Animal Care and Use Committee at the University of California, San Diego (Protocol # S08272) and according to National Institutes of Health and institutional guidelines.

## Decision letter and Author response
Decision letter https://doi.org/10.7554/eLife.42148.024
Author response https://doi.org/10.7554/eLife.42148.025

# Additional files

## Supplementary files
• Transparent reporting form
DOI: https://doi.org/10.7554/eLife.42148.018

## Data availability
MClust software is freely available from AD Redish at: http://redishlab.neuroscience.umn.edu/MClust/MClust.html. Chronux software is freely available at: http://chronux.org/. Source data files have been provided for Figure 4 and Figure 6. The data and custom Matlab analysis code that support the findings of this study are available from the Dryad Digital Repository: http://doi.org/10.5061/dryad.c89124p.

The following dataset was generated:

| Author(s) | Year | Dataset title | Dataset URL | Database and Identifier |
|---|---|---|---|---|
| Ewell LA, Fischer KB, Leibold C, Leutgeb S | 2019 | Data from: The impact of pathological high frequency oscillations on hippocampal network activity in rats with chronic epilepsy | https://doi.org/10.5061/dryad.c89124p | Dryad Digital Repository, 10.5061/dryad.c89124p |

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
