## [Decision Letter]

Thank you for submitting your article "The impact of pathological high frequency oscillations on hippocampal network activity in rats with chronic epilepsy" for consideration by *eLife*. Your article has been reviewed by three peer reviewers, one of whom is a member of our Board of Reviewing Editors, and the evaluation has been overseen by Laura Colgin as the Senior Editor. The reviewers have opted to remain anonymous.

The reviewers have discussed the reviews with one another and the Reviewing Editor has drafted this decision to help you prepare a revised submission.

Summary:

This manuscript from Ewell and Colleagues investigates normal and pathological high frequency oscillations (ripples vs. pHFOs, respectively) and their network effect on kainic acid rat model of temporal lobe epilepsy. The scope of the manuscript is important since it focuses on an electrophysiological signal (pHFOs) that is potentially considered as a biomarker of the epileptogenic zone.

Authors show several results but the most novel ones are that 1) PHFos can be observed during locomotion, which is never the case with normal ripples; 2) PHFos during locomotion decrease theta rhythm amplitude; 3) there is little overlap between neurons activated by pHFO and those activated during ripples; 4) PHFO occurrence decrease spatial information of hippocampal neurons that are in pHFO modulated, as well as instability and blurring of place fields. There is also some mechanistic investigation included, e.g. demonstration that interictal discharges/pHFOs disrupt hippocampal theta oscillations, and evidence that only some CA1 neurons are "hijacked" to participate in pHFOs.

The manuscript is clearly written and contains a trove of results that altogether make a very interesting paper. The methods and experimental design are sound, so are the interpretation of results. There are however a few points that deserve attention.

Essential revisions:

1) The authors must clearly highlight novel findings compared to previous studies, particularly from Dr. Menendez de la Prida's laboratory (Valero et al., 2017) on HFOs and ripple relationships and their contribution to cognition. This should be clearly stated in the Discussion.

2) pHFOs and interictal spikes are reported to often co-occur. This is well known (Jacobs et al., 2008; Salami et al., 2014 for example) and raises a specific point. To grasp the significance of the current results, it is important to know what the relative contribution of pHFOS vs. IS here. By themselves, IS have been found to alter ongoing oscillations, information processing and performance. Since pHFOs and IS often co-occur, it is possible that the effects reported on place cells and EEG are just due to the spikes and not the pHFO. For instance, it is not clear whether the disruption of theta activity can be observed when pHFOs occur without interictal spikes (see Kleen et al., 2010, 2013; Chauviere et al., 2009; Fu et al., 2018).

3) Related to this, HFOs can easily be confused with the filtered signal from interictal spikes. Indeed, interictal spike waveforms can be steep, leading to ripples in all frequency bands, including those that we focus on while detecting HFOs (see Benar et al., 2010). Restricting analysis to events with 6 cycles may not be sufficient if the filter used leads to ringing artifacts. The authors should describe the precautions taken to rule out filter artifacts. Please provide details of the filter used (type, order, frequency/amplitude response etc.), as ripple events could result from filtering population action potentials, and pHFOs could be a result of high-pass filtering a sharp wave. The authors allude to this issue in the first paragraph of the Discussion, but need to ensure this is not the case. Supplementary figures are helpful for this since they show unfiltered data, but authors need to comment on this in the main text.

4) A large chunk of the paper is devoted to the association between neural firing and pHFOs, which appear to be a proxy for interictal discharges in this model. The observation of sparse neuronal activity during high amplitude very high frequency events (up to 700 Hz, well into the range of multiunit activity) seems self-contradictory, and this needs to be resolved. There are some obvious possible explanations. First, extensive gliosis and cell death are well known to occur with the kainate model. Second, it is likely that the pHFOs are being recorded at a distance from the origination of the epileptiform activity. Finally, there is the possibility of filter artifact causing spurious pHFO detection (see point 3, above).

5) How did the authors guarantee that the same neurons were recorded across sessions? Changes in neuronal waveshape during epileptiform activity and over time have been reported in continuous recordings. Was there an analysis of waveshape drift that would support the contention that a given neuron was identified correctly across sessions? Also, please provide statistics for number of sessions of recording and total number of days from first to last recording per animal.

6) Please describe and appropriately cite how wake/sleep state is determined. Also, it is incorrect to state that pHFOs "do not occur during awake and REM sleep states." They are simply less common in some (not all) patients, and likely depend on the temporal pattern of interictal discharges which are often (not always) increased during sleep. Some of the papers cited by the authors provide data on pHFOs during wakefulness, as do several studies employing long-term, round the clock automated detection.

7) The authors should report test statistics, degrees of freedom, and actual p-values for statistical tests. It's not clear that their across-animal statistics were adequately powered. Some evidence of sufficient power would greatly support the early claims. Also, in some cases statistics are presented far removed from the accompanying claim, which is confusing to the reader. When dealing with multiple groups and ANOVA, could the authors provide the global statistical effect before comparing individual groups (CTR vs. ripple like, CTRL vs. pHFO etc.). Did authors adjust for multiple comparisons?

8) The conclusion that different neuronal populations could be selectively targeted ventures too far into speculation, as the authors did not show that the ripple vs. pHFO neurons came from different classes. This section should be removed or limited to a brief comment. Similarly, it is too much of a reach to propose that suppressing pHFOs would improve cognition, as the pHFOs are likely a manifestation of an interictal discharge and may also reflect other epilepsy-related abnormalities that contribute independently to the observed effects.

9) The supplementary information shows many individual examples from each recorded animal, and this inclusion extensive raw data is quite laudable. This could be improved by showing in panels A and B of Figure 2—figure supplement 1 single examples of each of the ripples, ripple-like events, and pHFOs.

10) It is striking that pHFOs occur all the time, and yet have relatively modest effects on behavior, and that they don't appear to "hijack" normal circuits in any reliable way. This might suggest that epileptic networks compensate in many ways, for example, to decrease activity of neurons contributing to pHFOs, or for networks to learn to develop firewalls serving to suppress the jumbled signal arising out of pathological activity. Such possibilities, while speculative, are clearly important questions for future research and should be further developed in the Discussion.

---

## [Author Response]

Essential revisions:1) The authors must clearly highlight novel findings compared to previous studies, particularly from Dr. Menendez de la Prida's laboratory (Valero et al., 2017) on HFOs and ripple relationships and their contribution to cognition. This should be clearly stated in the Discussion.

We thank the reviewers for providing the opportunity to better highlight the critical differences between our data and that of Valero et al. In the original version of our Discussion we cite Valero et al., 2017 in the context of highlighting previous work on how HFO and ripple affect cognition. However as noted by the reviewer, our original version did not clearly point out how our work reinforces/differs from this important previous work. There are two key differences between our data and that of Valero et al: we are both studying HFO, but in their case they exclude sessions with interictal spikes, and in our case we use the association with interictal spikes to define pHFO – therefore we are focusing on fundamentally distinct types of HFO. Furthermore, we track individual CA1 unit activity between HFOs associated with interictal spikes (pHFO) and those that are associated with sharp waves (ripple-like events), which they do not. The differential responses of units to these two HFO types support our novel claim that these high frequency events are fundamentally distinct. We have revised this paragraph to point out these differences, and more clearly compare our results (Discussion, fourth paragraph). We have also tried to better emphasize the novelty of our findings with respect to previous work throughout the Discussion.

2) pHFOs and interictal spikes are reported to often co-occur. This is well known (Jacobs et al., 2008; Salami et al., 2014 for example) and raises a specific point. To grasp the significance of the current results, it is important to know what the relative contribution of pHFOS vs. IS is here. By themselves, IS have been found to alter ongoing oscillations, information processing and performance. Since pHFOs and IS often co-occur, it is possible that the effects reported on place cells and EEG are just due to the spikes and not the pHFO. For instance, it is not clear whether the disruption of theta activity can be observed when pHFOs occur without interictal spikes (see Kleen et al., 2010, 2013; Chauviere et al., 2009; Fu et al., 2018).

We edited multiple sections of the manuscript to address these comments:

a) We now cite (Jacobs et al., 2008; Salami et al., 2014) when we first raise the concept that one population of HFO in our study co-occur with interictal spikes (subsection “Ripple-like oscillations and pathological high frequency oscillations occur in CA1 of the same animal”, second paragraph). Our original version cites Kleen et al., and we did not include Chauvier et al., 2009 because it did not address interictal spikes and theta. For Fu et al., 2018 see (d) below.

b) In our data we describe two types of HFO’s in the CA1 cell layer of epileptic rats that are classified as either associated with interictal spikes (pHFOs) or associated with normal size sharp waves (ripple-like). The first time the two events are defined, the association of pHFO with interictal spikes is stated:

“The other group, which we refer to as ‘pathological high frequency oscillations (pHFOs)’, were associated with large amplitude envelopes (interictal spikes) …”

For our recordings from the CA1 cell layer, pHFOs are thus defined as associated with interictal spikes, and vice versa, we do not observe interictal spikes without pHFOs at this recording location. However, simultaneously with the pHFO-interictal spike complex in the CA1 cell layer, we observe large interictal spikes without HFOs at recording locations in stratum radiatum (see also our response to Question 3 below). We believe these signatures in the cell layer and the radiatum to be components of the same event in the CA1 network (Author response image 1).

The co-occurrence of pHFOs and ictal spikes is stressed more frequently in this version of the manuscript. For example, a new supplemental figure (Figure 1—figure supplement 2) has been added that shows simultaneously recorded signal from the cell layer and stratum radiatum. Rhetorically we chose to refer to the pattern as pathological HFO (pHFO) because the event was detected based on frequency and much of our manuscript deals with comparisons of these events with normal HFO in the healthy brain and the second class of HFO in the epileptic brain (ripple-like) (5 of 7 main figures), in which only pHFOs differ in both amplitude and brain state dependence. Indeed, since pHFOs are the only class of HFOs in our study to regularly occur during theta states, they are the only HFO event with the potential to influence ongoing theta oscillations. However, to be clear, in our study pHFO and interictal spikes are two parts of one activity pattern, just as ripple and sharp waves are part of one activity pattern in the healthy hippocampus, i.e. ‘sharp wave ripples’.

c) With that said, we do think that more emphasis should be placed on the possible role of the interictal spike component in disrupting theta (given that we analyze theta effects from tetrodes positioned in radiatum where theta is stronger, and where interictal spikes dominated the interictal pattern). We reasoned that if the interictal spike were the key component, there might be a relationship between its magnitude and the strength of theta suppression. At our stratum radiatum recording sites, many events were saturated (Author response image 1), occluding their true magnitude. However, we found that the maximum slope of the IS was a good proxy for IS amplitude (Author response image 1). We found no relationship between IS slope (proxy for magnitude) and theta suppression. Furthermore we found no evidence for a relationship for any other parameter (IS amplitude in the cell layer, pHFO power, and pHFO frequency, Author response image 1). As this new analysis did not reveal any clear relation between pHFO/IS features and the degree of theta suppression, we have not added these results to the current version of manuscript but would of course include the additional analysis and figures if deemed critical by the reviewers and editor. Instead, we now more clearly make the point that the IS along with the associated HFOs in the cell layer could be an important factor for suppressing theta (subsection “pHFOs are not brain state dependent and occur during foraging epochs characterized by movement-related theta oscillations”, last paragraph).

**Author response image 1. respfig1:** No relationship between IS /pHFO parameters and magnitude of theta suppression. (**A**) Example event recorded simultaneously in the cell layer (stratum pyramidale) and radiatum during movement related theta oscillation. (**B**) In one foraging session, interictal spikes (IS) recorded in radiatum did not saturate and varied enough to allow us to test whether IS slope correlated with IS amplitude. The strong correlation allowed us to use slope (**C**) as a proxy for IS amplitude in other recording sessions where IS saturated (as shown in A). C,D,E No relationship was observed between radiatum IS slope (**C**), cell layer IS amplitude (**D**), or cell layer pHFO power/frequency (**E**) and magnitude of theta reduction. ID numbers refer to individual epileptic animals.

d) We thank the reviewer for pointing out the Fu et al., paper, and we now cite this work (see the aforementioned paragraph).

The Fu et al. paper shows interictal spikes that are followed by theta suppression. However, they sample at 256 Hz, and therefore it is unclear whether the interictal spikes in their study are associated with high frequency oscillations that could not be resolved or whether their recording locations were distant from a cell layer where HFOs would not be detected. We gather that this brings us full circle – the complexity of the distinction between interictal spike and HFO is likely underappreciated. It is our impression that recording location is fundamental, and without simultaneous recordings at multiple sites, the full extent of the co-occurring events may not be evident. Indeed, in our own experiments, if we had been recording in radiatum only, we would have falsely detected interictal spikes without associated pHFOs. While our manuscript does not indicate that one part of the complex is more important for disruption of normal network dynamics, it does highlight that in many cases, it is most parsimonious to think of these two patterns as a single event, at least within the hippocampus. We believe that emphasizing this result is a valuable and novel addition to our manuscript, and we appreciate that the reviewer comments pointed us in this direction.

3) Related to this, HFOs can easily be confused with the filtered signal from interictal spikes. Indeed, interictal spike waveforms can be steep, leading to ripples in all frequency bands, including those that we focus on while detecting HFOs (see Benar et al., 2010). Restricting analysis to events with 6 cycles may not be sufficient if the filter used leads to ringing artifacts. The authors should describe the precautions taken to rule out filter artifacts. Please provide details of the filter used (type, order, frequency/amplitude response etc.), as ripple events could result from filtering population action potentials, and pHFOs could be a result of high-pass filtering a sharp wave. The authors allude to this issue in the first paragraph of the Discussion, but need to ensure this is not the case. Supplementary figures are helpful for this since they show unfiltered data, but authors need to comment on this in the main text.

We thank the reviewer for asking us to include details about our HFO detection. This is fundamental to our study, and we are happy to provide these details (please note the detection code will also be published). We designed an Equiripple Bandpass Filter using the FIRPM function in the Signal Processing Toolbox of Matlab. The filter has an order of 477 (as we understand it, the order of digital filter is its length – please correct us if this is wrong). Our filter was designed to have the following features:

First stopband = 0 – 140 Hz, amplitude 0

Passband = 150 – 800 Hz, amplitude 1

Second stopband = 810 Hz – Nyquist Frequency, amplitude 0

The first and second stopband attenuation were set to 5.6 X 10^-4^. The passband attenuation was set to 2.8 X 10 ^-2^. These led to error minimization weights of 51.1689 and 1 respectively. In Author response image 2 we have plotted the magnitude response of the filter.

**Author response image 2. respfig2:** Magnitude response of the Equiripple Bandpass Filter.

The details of our filter have been added to the Materials and methods (subsection “Detection of high frequency oscillations”).

The reviewer is right to point out that great care needs to be taken to avoid misidentifying either interictal spikes or noise artifacts as HFOs. When we designed our detection algorithm, we were aware of these problems and took care to convince ourselves that our detection was not picking up false positives. First, for every session we visually evaluated the raw signals of candidate HFOs – we visualized at least 20 randomly selected events from each session. We have added this detail to the Materials and methods (see the aforementioned subsection). We found that in rare cases certain artifacts were detected as candidate HFO (large spikes or DC shifts were meeting the criteria – though this is rare, see below). Importantly, in those cases, our second criteria in which we calculated the fft of the raw signal eliminated these artifacts because they had high power between 75 – 125 Hz (see example Rejected events in Figure 1—figure supplement 1). Still, the point by the reviewer is important enough that we have added a second supplementary figure (Figure 1—figure supplement 2) to explore these issues more deeply. In our discussion of these data, we have also cited Benar et al. (subsection “Ripple-like oscillations and pathological high frequency oscillations occur in CA1 of the same animal”, first paragraph) to increase awareness of these important considerations when analysing HFO.

In our analysis, we detect HFO recorded on tetrodes located in the cell layer. This is where the HFO was strongest, and the concurrent interictal spike was smallest. In our experiments, other tetrodes located in stratum radiatum had simultaneously occurring interictal spikes that were larger (and unfortunately typically saturated). In one case, we have a recording session in which the simultaneously occurring interictal spikes in stratum radiatum were not saturating, and also had no visible HFO. This small data set serves as a well-suited control to test whether our HFO detection code would erroneously detect HFO when there are interictal spikes but no visible HFO. In Figure 1—figure supplement 2, we show examples of simultaneously recorded signal from the cell layer (A) and stratum radiatum (B). Despite the presence of large interictal spikes (> 2000 microvolts – even larger than spikes in the cell layer), no HFO were detected in the signal recorded from stratum radiatum. We also simulated troublesome artifacts such as large DC shifts (C) or fast biphasic artifacts (D). In both of these cases, our filter introduced some false oscillation into the signal, however there were not enough high powered cycles to meet our criteria of number of high power cycles (amplitude > 3 S.D). In preparing these figures, we determined that our original method description had an error. The cycle inclusion criteria for HFO is that it has at least 5, not 6, high power cycles. During initial testing we used 6, but found it was too stringent for control ripple detection. We apologize for this error and have corrected it in the Materials and methods and in Figure 1—figure supplement 1. We reran detection to verify that the variable was set to 5 for all recording sessions, which it was. In summary, in line with our previous validation, we have found that our filter is not introducing false oscillations in our data set. Interictal spikes are not causing oscillation artifact. Very fast artifacts cause some oscillatory artifact upon filtering, but in most cases the amplitudes are too small to reach criteria and in other cases the false positives were removed using further criteria based on an fft of the raw signal. These reasons combined with visual inspection of the raw signals give us confidence that our false positive rate is low.

4) A large chunk of the paper is devoted to the association between neural firing and pHFOs, which appear to be a proxy for interictal discharges in this model. The observation of sparse neuronal activity during high amplitude very high frequency events (up to 700 Hz, well into the range of multiunit activity) seems self-contradictory, and this needs to be resolved. There are some obvious possible explanations. First, extensive gliosis and cell death are well known to occur with the kainate model. Second, it is likely that the pHFOs are being recorded at a distance from the origination of the epileptiform activity. Finally, there is the possibility of filter artifact causing spurious pHFO detection (see point 3, above).

We may not have sufficiently clearly stated that the pHFOs we characterize in this study do not have peak frequencies above 320 Hz. We filtered with a much wider band (140 – 800 Hz), but we did not detect events with frequencies greater than 320 Hz. The frequency is reported in the Results, ‘pHFO n = 923, peak frequency median, 245.9 Hz; IQR, 233.7- 264.1 Hz’

Also please see Figure 2 and Figure 2—figure supplement 1. We wonder whether the frequencies could be mixed up with the amplitude of the envelope – which in the case of pHFOs are associated w/ interictal spikes that have amplitudes in the 700 microvolt range (so perhaps, this is where the 700 value could have been read). We revisited the sections in the Materials and methods and Results where the frequency range is described and did not find any obvious erroneous description.

Even though we were not able to trace the comment to a particular statement, the spirit of the comment is well taken – it is counterintuitive that unit activity would be sparse during high frequency events. We think that the most likely scenario is that the high frequency oscillation in the LFP corresponds to fast synaptic inhibition (similar to what is hypothesized for normal ripple). This is consistent with a report from Muldoon et al. (which we now cite, subsection “Sparse activation of CA1 principal cells during high frequency oscillations in animals with epilepsy”, first paragraph) that shows that interneurons are strongly activated during interictal activity.

5) How did the authors guarantee that the same neurons were recorded across sessions? Changes in neuronal waveshape during epileptiform activity and over time have been reported in continuous recordings. Was there an analysis of waveshape drift that would support the contention that a given neuron was identified correctly across sessions? Also, please provide statistics for number of sessions of recording and total number of days from first to last recording per animal.

We are happy to provide more detail about our manual cluster cutting practices. As mentioned by the reviewer, waveform shapes change during epileptiform activity, however in the current study neurons were never tracked across seizure events, and we found waveform shape to be robust against interictal activity. We did not track neurons across days – only within recording sessions which comprised resting and foraging behavior (Figure 1A). Our lab has extensive expertise in tracking the cell identities of individual hippocampal neurons within large populations across days (Mankin et al., 2012; Mankin et al., 2015, Neuron) and we have utilized the same methods and procedures for tracking individual neurons across recording sessions in our current submission. We have added an additional supplemental figure (Figure 4—figure supplement 1) which provides more detail of our cluster cutting, as well as generally reported cluster quality measures (L-Ratio and Isolation Distance) for neurons recorded in control and epileptic animals. Furthermore, per the reviewers request (and in our opinion, a good idea), we have included an analysis of waveform drift across recording sessions (Figure 4—figure supplement 1E, F) which shows that drift is minimal and not different between control and epileptic animals. We have added clarifying text to the Materials and methods section and details about recording session per animal (subsection “Cell sorting and cell-tracking”).

6) Please describe and appropriately cite how wake/sleep state is determined.

Thank you for the opportunity to clarify our language. In this study, we do not determine wake/sleep. Rather we compare epochs recorded in different environments which promote different types of behavior. During foraging epochs, animals are well trained to run continuously and we use running speed to qualify behavioral brain state. During resting epochs, animals are in a small comfortable container – we can only say that animals are comparatively immobile – however we do not distinguish between awake immobile and sleeping. In normal rodents, ripples occur in both of these restful states, so we did not think it necessary to delineate between the two for this study. We have gone through the text to make sure we are clear in our description of the animal’s behavior.

Also, it is incorrect to state that pHFOs "do not occur during awake and REM sleep states." They are simply less common in some (not all) patients, and likely depend on the temporal pattern of interictal discharges which are often (not always) increased during sleep. Some of the papers cited by the authors provide data on pHFOs during wakefulness, as do several studies employing long-term, round the clock automated detection.

We apologize to this over simplification and have changed this sentence to read

‘Contrary to our result, there are reports that fast oscillations in epilepsy are more rare during awake and REM sleep states and instead occur predominantly during non-REM sleep (Bagshaw et al., 2009; Frauscher et al., 2015).’7) The authors should report test statistics, degrees of freedom, and actual p-values for statistical tests. It's not clear that their across-animal statistics were adequately powered. Some evidence of sufficient power would greatly support the early claims. Also, in some cases statistics are presented far removed from the accompanying claim, which is confusing to the reader. When dealing with multiple groups and ANOVA, could the authors provide the global statistical effect before comparing individual groups (CTR vs. ripple like, CTRL vs. pHFO etc.). Did authors adjust for multiple comparisons?

We now report exact p-values, test statistics, and where relevant, degrees of freedom. Furthermore, when dealing with multiple groups we now provide global statistics followed by a description of post hoc multiple comparisons. We have adjusted for multiple comparison – and state the post hoc test in the text where relevant.

8) The conclusion that different neuronal populations could be selectively targeted ventures too far into speculation, as the authors did not show that the ripple vs. pHFO neurons came from different classes. This section should be removed or limited to a brief comment.

We have removed the comment about selective targeting, but we think that the discussion of possible different classes participating in ripple-like versus pHFO warrants discussion, given the knowledge that in the normal brain only superficial CA1 cells are thought to be active during ripple. Furthermore, we have shifted the emphasis of intervention away from targeting based on cell-type specificity, and instead raised the point that pHFOs could be selectively manipulated using closed-loop approaches that don’t depend on cellular identity, but take advantage of the fact that different networks are involved in the two oscillation types (Discussion, last paragraph).

Similarly, it is too much of a reach to propose that suppressing pHFOs would improve cognition, as the pHFOs are likely a manifestation of an interictal discharge and may also reflect other epilepsy-related abnormalities that contribute independently to the observed effects.

This conclusion has been altered in the discussion to highlight that ripple-like events are maintained in epileptic networks, which highlights the possibility for successful therapy (i.e. there is something normal-like left). Further, the network dynamics during interictal events (pHFOs/spikes) are segregated from network dynamics during ripple-like events. We believe that pointing out that this segregation allows for intervention that selectively targets pathological activity and speculating whether this could be beneficial is appropriate for the Discussion (Discussion, last paragraph). We believe a strength of our manuscript is a careful description of hippocampal network mechanisms that are generally impaired in the epileptic brain, versus acute effects of pHFO to ongoing information processing. We have added text highlighting the subtle effects that pHFO had on ongoing network coding in our study (point 10 below) which puts our discussion of intervention in the proper context (Discussion, sixth paragraph). However, we do raise the point that long-term effects of pHFO frequency on hippocampal connectivity, function and cognition is an important area of research needed for motivating the use of long-term pHFO suppression for therapeutic intervention.

9) The supplementary information shows many individual examples from each recorded animal, and this inclusion extensive raw data is quite laudable. This could be improved by showing in panels A and B of Figure 2—figure supplement 1 single examples of each of the ripples, ripple-like events, and pHFOs.

We are thankful that our effort to include ample amounts of raw data has been appreciated. In response to the point raised we have added examples of each event type as panels A – C of Figure 2—figure supplement 1. This new inclusion is extremely helpful for visualizing how the combination of the frequency of the high frequency oscillation and the amplitude of the envelope (voltage deflection between 0.2 – 40 Hz) distinguishes pathological from ripple-like events.

10) It is striking that pHFOs occur all the time, and yet have relatively modest effects on behavior, and that they don't appear to "hijack" normal circuits in any reliable way. This might suggest that epileptic networks compensate in many ways, for example, to decrease activity of neurons contributing to pHFOs, or for networks to learn to develop firewalls serving to suppress the jumbled signal arising out of pathological activity. Such possibilities, while speculative, are clearly important questions for future research and should be further developed in the Discussion.

We agree that it is important to stress the relatively modest acute effects on behavior, and have added a few sentences to the discussion (Discussion, sixth paragraph). We have kept the addition short as our discussion was lengthened by adding a direct comparison with the Valero paper (Major point 1).